# Support-Proximity Augmented Diffusion Estimation
# for Offline Black-Box Optimization

Yonghan Yang [* 1]   Ye Yuan [* † 1 2 3]   Zipeng Sun [2 3]   Linfeng Du [2 3]
Bowei He [1]   Haolun Wu [2 3]   Can Chen [3 4 ‡]   Xue Liu [1 2 3]

## Abstract

Offline black-box optimization aims to discover novel designs with high property scores using only a static dataset, a task fundamentally challenged by the out-of-distribution (OOD) extrapolation problem. Existing approaches typically bifurcate into inverse methods, which struggle with the ill-posed nature of mapping scores to designs, and forward methods, which often lack the distributional expressivity to quantify uncertainty effectively. In this work, we propose **SPADE** (**S**upport-**P**roximity **A**ugmented **D**iffusion **E**stimation), a novel framework that reimagines forward surrogate modeling through the lens of conditional generative modeling. SPADE models the forward likelihood $p(y|\boldsymbol{x})$ using a diffusion model, but with two critical enhancements to tailor it for optimization: (1) a *Calibrated Diffusion Estimation* module that enforces global consistency in statistical moments and pairwise rankings, and (2) a *Support-Proximity Regularization* mechanism that implicitly internalizes the data manifold constraint $p(\boldsymbol{x})$ via kNN-based density estimation. Theoretically, we prove that our regularization is first-order equivalent to maximizing a Bayesian posterior with a valid design prior. Empirically, SPADE achieves state-of-the-art performance across Design-Bench tasks and an LLM data mixture optimization benchmark. Our code is available at https://github.com/HarryYoung2018/spade.

## 1. Introduction

The problem of discovering optimal designs from static datasets, known as offline black-box optimization (BBO) (Trabucco et al., 2022; Kim et al., 2026), is central to numerous scientific and engineering disciplines where online data collection is prohibitively expensive or dangerous. Applications range from designing biological sequences such as proteins (Sample et al., 2019) and DNA (Barrera et al., 2016; Le et al., 2018; Brookes et al., 2019) to optimizing hardware accelerators (Xue et al., 2025) and tuning robot controllers (Brockman et al., 2016; Ahn et al., 2020). Unlike active learning settings, the offline learner must infer the objective landscape and identify high-performing candidates solely from fixed historical data, without access to the ground-truth oracle for verification. Consequently, the core challenge lies in the *out-of-distribution (OOD) problem*: optimization algorithms naturally tend to exploit the epistemic errors of the surrogate model, gravitating towards adversarial regions where performance is overestimated.

To tackle this, the literature has largely diverged into two paradigms. *Inverse methods* directly model the conditional design distribution $p(\boldsymbol{x}|y)$ using generative models, allowing for the generation of candidates conditioned on high target scores (Krishnamoorthy et al., 2023; Yuan et al., 2025a). While appealing, learning the inverse mapping is often an ill-posed one-to-many problem, making these models difficult to train and prone to mode collapse. Conversely, *forward methods* learn a surrogate $y \approx f_{\boldsymbol{\theta}}(\boldsymbol{x})$ to guide the search (Trabucco et al., 2021; Yuan et al., 2023). However, standard regression models (e.g., deterministic neural networks) often fail to capture the complex distributional uncertainty required to detect OOD risks, while ensemble-based or Bayesian neural networks can be computationally intensive and hard to scale.

We identify three key limitations in current methodologies. First, while diffusion models have revolutionized inverse approaches to capture $p(\boldsymbol{x}|y)$, their potential as robust forward surrogates for modeling $p(y|\boldsymbol{x})$ remains *underexplored*. Forward modeling is generally a better-posed many-to-one problem, yet standard regression baselines lack the probabilistic flexibility of diffusion models to capture heteroscedasticity

*Equal contribution with random order: Ye designs algorithm/drafts paper; Yonghan conducts experiments. †This work is done during Ye's visit at MBZUAI. ‡Work done independent of the author's position at Amazon AGI. [1]MBZUAI - Mohamed bin Zayed University of Artificial Intelligence [2]McGill University [3]Mila - Quebec AI Institute [4]Amazon AGI. Correspondence to: Ye Yuan <ye.yuan3@mail.mcgill.ca>.

*Proceedings of the 43rd International Conference on Machine Learning*, Seoul, South Korea. PMLR 306, 2026. Copyright 2026 by the author(s).

and multi-modality. Second, simply replacing a regressor with a diffusion model is insufficient. Standard diffusion training focuses on capturing the training distribution, which does not guarantee calibrated expected values or global ranking accuracy required for effective optimization. A naive application of diffusion models may fail to serve as a reliable surrogate. Third, a standalone forward likelihood $p(y|\boldsymbol{x})$ ignores the prior constraint $p(\boldsymbol{x})$. Without an explicit mechanism to penalize low-density regions, even a probabilistic surrogate can be exploited by the optimizer in unsupported areas far from the data manifold.

Motivated by these insights, we propose **SPADE** (**S**upport-**P**roximity **A**ugmented **D**iffusion **E**stimation), a unified framework that constructs a robust, uncertainty-aware surrogate for offline optimization. As illustrated in Figure 1, SPADE addresses the aforementioned gaps through a holistic workflow that enhances standard diffusion training with two synergistic modules. To ensure the diffusion model serves as an effective surrogate, we introduce *Calibrated Diffusion Estimation*, which regularizes the training with first-order moment matching and rank consistency losses, aligning the surrogate's predictions with the global landscape topology. To incorporate prior constraints, we introduce *Support-Proximity Regularization*. This mechanism leverages a non-parametric $k$-nearest neighbor ($k$NN) estimator to measure data support and explicitly penalizes the surrogate's predictions via mean shrinking and variance inflation in OOD regions. Crucially, we prove that this geometric regularization is theoretically equivalent to optimizing the Bayesian posterior $p(\boldsymbol{x}|y) \propto p(y|\boldsymbol{x})p(\boldsymbol{x})$.

Our contributions are summarized as follows:

- We explore the potential of diffusion models for *forward surrogate modeling*, demonstrating their superiority over deterministic regression in capturing complex predictive distributions for offline BBO.

- We introduce a *Calibrated Diffusion Estimation* module incorporating first-order moment matching and pairwise ranking regularization, effectively tailoring the diffusion training objective to the requirements of optimization tasks.

- We propose *Support-Proximity Regularization* to enforce data manifold constraints and provide a theoretical proof that it is first-order equivalent to maximizing the joint objective of utility and prior probability.

- We conduct experiments on Design-Bench and an LLM data mixture optimization benchmark, showing that SPADE achieves state-of-the-art performance.

**Conflict of Interest Disclosure.** All authors are affiliated with academic institutions. The benchmarks and baselines

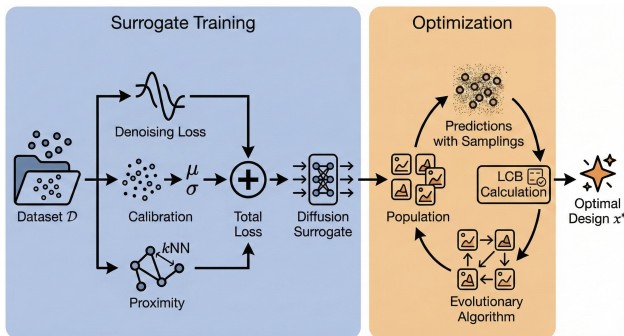

*Figure 1.* The pipeline consists of two stages: **Surrogate Training** learns a conditional diffusion model regularized by *Calibrated Diffusion Estimation* (ensuring moment and rank consistency) and *Support-Proximity Regularization* (penalizing OOD regions via kNN distances). **Optimization** then searches for the optimal design $\boldsymbol{x}^*$ using an Evolutionary Algorithm to maximize the Lower Confidence Bound (LCB) derived from the surrogate.

evaluated in this work were not developed by any author's employer, and the authors declare no financial conflicts of interest related to this work.

## 2. Preliminary

### 2.1. Offline Black-Box Optimization

Offline black-box optimization (BBO) studies the problem of discovering a design $\boldsymbol{x}^* \in \mathcal{X} \subseteq \mathbb{R}^D$ that maximizes an unknown objective function $f : \mathcal{X} \to \mathbb{R}$, given access only to a static dataset of past evaluations $\mathcal{D} = \{(\boldsymbol{x}_i, y_i)\}_{i=1}^N$, where $\boldsymbol{x}_i$ denotes an existing design and $y_i = f(\boldsymbol{x}_i)$ represents its corresponding property score (Trabucco et al., 2022; Kim et al., 2026). Unlike online optimization or active learning, the learner must rely solely on $\mathcal{D}$ to infer both the structure of the objective function and an appropriate notion of uncertainty for extrapolation. No additional evaluations of $f$ are available during training or optimization.

A unified probabilistic formulation of the offline optimization problem is given by the joint distribution of designs and property scores, $p(\boldsymbol{x}, y)$. By Bayes' rule, the conditional distribution of designs given a property score can be expressed as (Kim et al., 2026):

$$p(\boldsymbol{x} \mid y) \;\propto\; p(\boldsymbol{x})\, p(y \mid \boldsymbol{x}), \qquad (1)$$

where $p(y \mid \boldsymbol{x})$ models the forward relationship between design and property, and $p(\boldsymbol{x})$ serves as a prior over feasible or data-supported regions of $\mathcal{X}$. This decomposition clarifies the connection between two major modeling paradigms in offline BBO. *Forward methods* approximate $p(y \mid \boldsymbol{x})$ to predict outcome scores (Trabucco et al., 2021; Yuan et al., 2023). *Inverse methods*, in contrast, learn $p(\boldsymbol{x} \mid y)$ directly to generate new designs conditioned on desired property scores (Brookes et al., 2019; Krishnamoorthy et al., 2023).

## 2.2. Optimization Strategies

Once a forward surrogate is learned, the search for $\boldsymbol{x}^*$ requires an optimization strategy. The most prevalent approach in existing literature involves performing *gradient ascent* directly on a deterministic surrogate $f_\theta(\boldsymbol{x})$ (Trabucco et al., 2021; Yuan et al., 2023). Starting from an initial design $\boldsymbol{x}_0 \in \mathcal{D}$, the design is iteratively updated via $\boldsymbol{x}_{t+1} = \boldsymbol{x}_t + \eta \nabla_{\boldsymbol{x}} f_\theta(\boldsymbol{x}_t)$. While straightforward, this method is inherently local and prone to getting stuck in local optima or exploiting adversarial regions where the deterministic model overestimates performance without warning.

An alternative strategy, widely used in Bayesian Optimization, is to maximize an *acquisition function* $\mathcal{A}(\boldsymbol{x})$. Constructed from a probabilistic surrogate that outputs a distribution $\mathcal{N}(\mu(\boldsymbol{x}), \sigma^2(\boldsymbol{x}))$, the acquisition function $\mathcal{A}(\mu, \sigma)$ (e.g., Lower Confidence Bound) quantifies the utility of a candidate by incorporating both the expected performance ($\mu$) and the epistemic uncertainty ($\sigma$). However, this powerful paradigm remains underutilized in offline BBO because standard surrogates (e.g., MLPs) are deterministic and fail to provide the uncertainty $\sigma(\boldsymbol{x})$ necessary for risk-aware acquisition. This limitation necessitates the use of probabilistic models, such as Gaussian Processes or our proposed Diffusion framework, to enable robust global search via acquisition maximization.

## 2.3. Kernel and kNN Density Estimation

Kernel density estimation (KDE) provides a nonparametric approach to approximate the underlying data distribution $p(\boldsymbol{x})$. Given a set of samples $\{\boldsymbol{x}_i\}_{i=1}^N$ drawn from an unknown distribution and a kernel function $K_h(\cdot)$ with bandwidth parameter $h > 0$, the KDE of $p(\boldsymbol{x})$ is defined as:

$$\hat{p}_{\text{kde}}(\boldsymbol{x}) = \frac{1}{Nh^D} \sum_{i=1}^N K\left(\frac{\|\boldsymbol{x} - \boldsymbol{x}_i\|}{h}\right), \qquad (2)$$

where $K(\cdot)$ is a symmetric, positive kernel.

A widely used estimator is the $k$-nearest neighbor ($k$NN) density estimator, which can be viewed as a special case of KDE with an adaptive bandwidth determined by the distance to the $k$-th nearest neighbor and a uniform kernel $K(u) \propto \mathbb{I}\{\|u\| \leq 1\}$ (Gao et al., 2017) in Eq. (2). Specifically, let $R_k(\boldsymbol{x})$ denote the Euclidean distance from $\boldsymbol{x}$ to its $k$-th nearest neighbor among the $N$ samples, and let $V_D = \pi^{D/2}/\Gamma(\frac{D}{2} + 1)$ denote the volume of the $D$-dimensional unit ball. The kNN density estimator is then defined as:

$$\hat{p}_{\text{knn}}(\boldsymbol{x}) = \frac{k}{N V_D R_k(\boldsymbol{x})^D}. \qquad (3)$$

The kNN estimator adapts its local bandwidth according to the sample density, offering robustness in high-dimensional or irregularly distributed datasets.

---

**Algorithm 1 SPADE**

1: **Input:** Static dataset $\mathcal{D}$, regularization weights $\lambda_1, \lambda_2$, LCB coefficient $\beta$, number of neighbors $k$.
2: **Output:** Optimal design $\boldsymbol{x}^*$.
3: /* Surrogate Training Stage */
4: Initialize network $\epsilon_{\boldsymbol{\theta}}(\cdot)$ and build $k$NN index on $\mathcal{D}$.
5: **while** not converged **do**
6:     Sample mini-batch $\mathcal{B} = \{(\boldsymbol{x}, y)\} \sim \mathcal{D}$.
7:     /* Calibrated Diffusion Estimation */
8:     Compute base denoising loss $\mathcal{L}_{\text{diff}}$ using Eq. (6).
9:     Generate short-run MC samples for $\boldsymbol{x} \in \mathcal{B}$ to estimate $\hat{\mu}_\theta(\boldsymbol{x}), \hat{\sigma}_\theta(\boldsymbol{x})$.
10:     Compute calibration loss $\mathcal{L}_{\text{calib}}(\theta; \mathcal{B})$ using Eq. (7).
11:     /* Support-Proximity Regularization */
12:     Query $k$NN to get distance $d(\boldsymbol{x})$ and neighbor mean $\mu_{\text{NN}}(\boldsymbol{x})$.
13:     Compute proximity loss $\mathcal{L}_{\text{prox}}(\theta; \mathcal{B})$ using Eq. (8).
14:     Update $\boldsymbol{\theta}$ to minimize total loss $\mathcal{L}$ in Eq. (10).
15: **end while**
16: /* Optimization Stage */
17: Initialize population $\mathcal{P}$ with high-scoring designs.
18: **for** $gen = 1, 2, \cdots, G$ **do**
19:     **for** candidate $\boldsymbol{x} \in \mathcal{P}$ **do**
20:         Generate $M$ samples $y^{(m)} \sim p_{\boldsymbol{\theta}^*}(\cdot|\boldsymbol{x})$.
21:         Calculate $\text{LCB}(\boldsymbol{x})$ using Eq. (12).
22:     **end for**
23:     Update $\mathcal{P}$ using Evolutionary Algorithm.
24: **end for**
25: $\boldsymbol{x}^* \longleftarrow \arg\max_{\boldsymbol{x} \in \mathcal{P}} \text{LCB}(\boldsymbol{x})$
26: **return** $\boldsymbol{x}^*$

---

## 3. Methodology

Guided by the unified probabilistic formulation in Eq. (1), we aim to discover optimal designs by modeling the joint probability of high scores and feasible data support. To this end, we propose **SPADE** (**S**upport-**P**roximity **A**ugmented **D**iffusion **E**stimation). SPADE is composed of two synergistic modules. We begin in Section 3.1 with (1) **Calibrated Diffusion Estimation**, establishing a conditional diffusion surrogate $p_\theta(y \mid \boldsymbol{x})$ that is explicitly regularized for accurate statistical moments and pairwise rankings. Complementing this, Section 3.2 introduces (2) **Support-Proximity Regularization**, a mechanism that penalizes predictions in low-density regions via kNN proximity, thereby internalizing the design prior $p(\boldsymbol{x})$ into the likelihood model. Finally, in Section 3.3, we describe the joint training objective and the acquisition-based optimization strategy. The pseudo-code of the entire framework is provided in Algorithm 1.

### 3.1. Calibrated Diffusion Estimation

The foundation of our framework is a forward surrogate that models the conditional distribution of property scores $p(y \mid \boldsymbol{x})$. As highlighted in Section 2.2, effective acquisition-

based optimization requires quantifying epistemic uncertainty. While standard regression models yield deterministic point estimates, they fail to provide the distributional information necessary for such risk-aware search. To address this, we employ a Denoising Diffusion Probabilistic Model (DDPM) (Ho et al., 2020) to capture the full predictive distribution of the property scores.

**Base Diffusion Surrogate.** We parameterize $p_\theta(y \mid \boldsymbol{x})$ using a conditional diffusion process. Let $\{\beta_t\}_{t=1}^T$ be a fixed variance schedule, define $\alpha_t = 1 - \beta_t$, and $\bar{\alpha}_t = \prod_{s=1}^t \alpha_s$. The forward process diffuses a clean property score $y_0$ into Gaussian noise: $q(y_t \mid y_0) = \mathcal{N}(\sqrt{\bar{\alpha}_t}\, y_0,\, (1 - \bar{\alpha}_t)\mathbf{I})$. The reverse generative process is parameterized by a noise-prediction network $\epsilon_\theta$ conditioned on the design $\boldsymbol{x}$:

$$p_\theta(y_{t-1} \mid y_t, \boldsymbol{x}) = \mathcal{N}\Big(\mu_\theta(y_t, t, \boldsymbol{x}),\, \tilde{\beta}_t \mathbf{I}\Big), \quad (4)$$

$$\mu_\theta(y_t, t, \boldsymbol{x}) = \frac{1}{\sqrt{\alpha_t}}\left(y_t - \frac{1 - \alpha_t}{\sqrt{1 - \bar{\alpha}_t}}\, \epsilon_\theta(y_t, t, \boldsymbol{x})\right), \quad (5)$$

where $\tilde{\beta}_t$ represents the variance of the reverse step. We train $\epsilon_\theta$ using the standard denoising score matching objective:

$$\mathcal{L}_{\text{diff}}(\theta) = \mathbb{E}_{(\boldsymbol{x}, y) \sim \mathcal{D}}\, \mathbb{E}_{t, \epsilon} \big\| \epsilon - \epsilon_\theta(y_t, t, \boldsymbol{x}) \big\|_2^2, \quad (6)$$

where $y_t$ is the noisy version of $y$ at step $t$.

**Calibration via Moment and Rank Regularization.** While the denoising loss in Eq. (6) effectively captures the underlying conditional distribution, downstream optimization relies heavily on the *global statistics* and *relative ordering* of the candidates. Specifically, the optimizer utilizes the expected value $\hat{\mu}_\theta(\boldsymbol{x}) = \mathbb{E}_{p_\theta}[y]$ to estimate utility, and the pairwise rankings to distinguish superior designs from inferior ones. However, the standard objective operates at the level of noise prediction and does not explicitly guarantee accuracy in these global metrics. To bridge this gap, we introduce a *calibration loss* computed on a mini-batch $\mathcal{B}$ sampled from $\mathcal{D}$. For each $\boldsymbol{x} \in \mathcal{B}$, we explicitly estimate the predictive mean $\hat{\mu}_\theta(\boldsymbol{x}) \approx \frac{1}{M} \sum_{m=1}^M y^{(m)}$ by generating $M$ short-run Monte Carlo (MC) samples $y^{(m)} \sim p_\theta(\cdot \mid \boldsymbol{x})$:

$$\mathcal{L}_{\text{calib}}(\theta; \mathcal{B}) = \underbrace{\frac{1}{|\mathcal{B}|} \sum_{(\boldsymbol{x}, y) \in \mathcal{B}} \big(\hat{\mu}_\theta(\boldsymbol{x}) - y\big)^2}_{\text{1st-order moment matching}}$$

$$+ \underbrace{\frac{1}{|\mathcal{P}|} \sum_{(i,j) \in \mathcal{P}} \log\big(1 + \exp\{-s[\hat{\mu}_\theta(\boldsymbol{x}_i) - \hat{\mu}_\theta(\boldsymbol{x}_j)]\}\big)}_{\text{rank consistency}}, \quad (7)$$

where $\mathcal{P} = \{(i,j) \in \mathcal{B} \times \mathcal{B} \mid y_i > y_j\}$ denotes the set of strictly ordered pairs in the mini-batch, and $s > 0$ is a temperature scaling factor. The first term anchors the predictive mean to the ground truth, while the second term

enforces ranking consistency, ensuring that the surrogate's topology aligns with the true objective landscape. We set $s = 1$ in our experiments across all tasks.

### 3.2. Support-Proximity Regularization

A calibrated surrogate is necessary but insufficient for offline BBO due to the OOD problem: optimizers invariably exploit regions where the model overestimates performance. To mitigate this, we must enforce the prior constraint $p(\boldsymbol{x})$. Instead of training a separate generative model for $p(\boldsymbol{x})$, which is computationally expensive and hard to tune, SPADE introduces a geometric regularization mechanism that effectively internalizes this prior into the surrogate's predictions.

**Measuring Support Proximity.** We utilize the $k$-nearest neighbor (kNN) distance as a robust, non-parametric proxy for the support density. For a query $\boldsymbol{x}$, let $R_k(\boldsymbol{x})$ denote the Euclidean distance to its $k$-th nearest neighbor in $\mathcal{D}$. From Eq. (3), the negative log-density is proportional to the log-distance: $-\log \hat{p}_{\text{knn}}(\boldsymbol{x}) \propto d(\boldsymbol{x})$, where $d(\boldsymbol{x}) = \log R_k(\boldsymbol{x})$. Designs with large $d(\boldsymbol{x})$ are far from the data manifold and thus unreliable.

**Proximity-Based Mean Shrinking and Variance Inflation.** We propose *Support-Proximity Regularization* to penalize the model in low-density regions. Intuitively, for OOD designs, we force the surrogate to be *conservative*: predicting a lower mean and a higher uncertainty. For a mini-batch $\mathcal{B}$, utilizing the sample statistics $\hat{\mu}_\theta(\boldsymbol{x})$ and $\hat{\sigma}_\theta(\boldsymbol{x})$ estimated from $M$ MC draws, the loss is defined as:

$$\mathcal{L}_{\text{prox}}(\theta; \mathcal{B}) = \frac{1}{|\mathcal{B}|} \sum_{\boldsymbol{x} \in \mathcal{B}}$$
$$\Big[ \underbrace{\max\big(0,\, \hat{\mu}_\theta(\boldsymbol{x}) - \mu_{\text{NN}}(\boldsymbol{x}) - \tau(d(\boldsymbol{x}))\big)}_{\text{mean-shrink}} \quad (8)$$
$$+ \underbrace{\max\big(0,\, \sigma_{\min}(d(\boldsymbol{x})) - \hat{\sigma}_\theta(\boldsymbol{x})\big)}_{\text{variance-floor}} \Big],$$

where $\mu_{\text{NN}}(\boldsymbol{x})$ is the average score of neighbors, $\tau(d) = a\,d$ is a distance-dependent penalty margin, and $\sigma_{\min}(d) = a_0 + a_1 d$ sets a distance-dependent uncertainty floor. Unless otherwise specified, we fix the hyperparameters to $a = 0.02$, $a_0 = 0.02$, and $a_1 = 0.005$ across all experiments without task-specific tuning.

**Theoretical Analysis.** As discussed in Section 2.2, acquisition functions $\mathcal{A}(\mu, \sigma)$ play a pivotal role in guiding the optimization. Our proposed regularization explicitly aligns this acquisition process with the unified probabilistic view in Eq. (1). We establish this connection through the following proposition:

**Proposition 3.1** (First-order Equivalence to Prior Augmentation). *Let $\mathcal{A}(\mu, \sigma)$ be a continuously differentiable acqui-*

*sition function that is strictly increasing in $\mu$ and strictly decreasing in $\sigma$, i.e., $\partial_\mu \mathcal{A} > 0$ and $\partial_\sigma \mathcal{A} < 0$.*

*For a design $\boldsymbol{x}$, write $\mu = \hat{\mu}_\theta(\boldsymbol{x})$ and $\sigma = \hat{\sigma}_\theta(\boldsymbol{x})$, and define the support distance $d(\boldsymbol{x}) = \log R_k(\boldsymbol{x})$.*

***Support transform:***

$$T_d : (\mu, \sigma) \mapsto \big(\mu - \tau(d), \max\{\sigma, \sigma_{\min}(d)\}\big),$$
$$\tau(d) = a\,d, \quad \sigma_{\min}(d) = a_0 + a_1 d,$$

*with $a, a_0, a_1 \geq 0$.*

***Support-aware acquisition:***

$$\widetilde{\mathcal{A}}(\boldsymbol{x}) = \mathcal{A}\big(T_{d(\boldsymbol{x})}(\mu, \sigma)\big).$$

*Then there exist constants $C(\boldsymbol{x}) \in \mathbb{R}$ and a coefficient $\kappa(\boldsymbol{x})$ given by:*

$$\kappa(\boldsymbol{x}) = \frac{1}{D}\Big(a\,\partial_\mu \mathcal{A}(\mu, \sigma) - a_1\,\partial_\sigma \mathcal{A}(\mu, \sigma)\,\mathbb{I}\{\sigma < a_0\}\Big) > 0,$$

*where $D$ is the dimension of designs.*

*As $d(\boldsymbol{x}) \to 0$, noticing that $d(\boldsymbol{x}) \propto -\log \hat{p}_{knn}(\boldsymbol{x})$, we have:*

$$\begin{aligned} \widetilde{\mathcal{A}}(\boldsymbol{x}) = \mathcal{A}(\mu, \sigma) + \kappa(\boldsymbol{x}) \log \hat{p}_{knn}(\boldsymbol{x}) \\ + C(\boldsymbol{x}) + o\big(\log \hat{p}_{knn}(\boldsymbol{x})\big). \end{aligned} \quad (9)$$

***Consequently, maximizing the support-aware acquisition $\widetilde{\mathcal{A}}(\boldsymbol{x})$ is, to first order, equivalent to maximizing the joint objective of utility and prior probability:***

$$\underbrace{\mathcal{A}(\mu, \sigma)}_{Likelihood\ Utility} + \underbrace{\kappa(\boldsymbol{x}) \log \hat{p}_{knn}(\boldsymbol{x})}_{Prior\ Constraint}.$$

This confirms that SPADE effectively bridges the gap between forward modeling and Bayesian inference: Section 3.1 maximizes the likelihood $p(y|\boldsymbol{x})$, while Section 3.2 injects the log-prior $\log p(\boldsymbol{x})$, collectively optimizing the posterior target in Eq. (1). We emphasize that Proposition 3.1 serves a theoretical motivation for the regularization design rather than a direct justification of the full algorithm's behavior. We provide the detailed proof in Appendix A.

### 3.3. Training Objective and Inference

**Training Objective.** We jointly train the diffusion surrogate by minimizing the total objective, which combines the base denoising loss with the proposed calibration and proximity regularizations:

$$\mathcal{L}(\theta) = \mathcal{L}_{\mathrm{diff}}(\theta) + \lambda_1\,\mathcal{L}_{\mathrm{calib}}(\theta; \mathcal{B}) + \lambda_2\,\mathcal{L}_{\mathrm{prox}}(\theta; \mathcal{B}), \quad (10)$$

where $\lambda_1$ and $\lambda_2$ are hyperparameters controlling the strength of regularizations.

**Inference and Optimization.** Unlike deterministic methods, our probabilistic surrogate enables global search via acquisition maximization. To strictly enforce conservatism against OOD risks, we employ the Lower Confidence Bound (LCB) as the acquisition function. Formally, given a hyperparameter $\beta > 0$, the LCB of a candidate $\boldsymbol{x}$ is estimated using $M$ Monte Carlo samples:

$$\mathrm{LCB}(\boldsymbol{x}) = \hat{\mu}_\theta(\boldsymbol{x}) - \beta \hat{\sigma}_\theta(\boldsymbol{x}) \quad (11)$$

$$\approx \frac{1}{M}\sum_{m=1}^{M} y^{(m)} - \beta \sqrt{\frac{1}{M-1}\sum_{m=1}^{M}(y^{(m)} - \bar{y})^2}, \quad (12)$$

where $y^{(m)} \sim p_\theta(\cdot \mid \boldsymbol{x})$, and $\bar{y}$ is the mean of the $M$ Monte Carlo samples. Crucially, thanks to the support-proximity regularization ($\mathcal{L}_{\mathrm{prox}}$), OOD candidates naturally yield lower LCB values due to suppressed means and inflated variances, effectively guiding the search towards high-density regions. To maximize Eq. (12), we employ a standard evolutionary algorithm (EA) initialized with high-scoring seeds from $\mathcal{D}$.

## 4. Experiments

In this section, we empirically evaluate SPADE to answer three primary research questions. First, **RQ1** investigates whether SPADE outperforms existing state-of-the-art offline black-box optimization methods in discovering high-scoring designs. Second, **RQ2** examines the individual contributions of the proposed *Calibrated Diffusion Estimation* and *Support-Proximity Regularization* modules to the overall performance. Third, **RQ3** explores the robustness of our framework with respect to the choice of optimization strategy, specifically testing whether SPADE remains effective when the default Lower Confidence Bound (LCB) is replaced with other acquisition functions.

To answer these questions, we structure our analysis as follows. We begin by detailing the experimental setup and baselines in Section 4.1. In Section 4.2, we present a comprehensive quantitative evaluation on the Design-Bench suite and an LLM data mixture optimization task to address **RQ1**. We then address **RQ2** through ablation studies isolating each component in Section 4.3. Finally, results for **RQ3**, along with computational cost analyses and qualitative visualizations on synthetic 2D functions, are detailed in Appendices B, C, and D.

### 4.1. Experimental Setup

**Datasets.** We evaluate SPADE on a diverse suite of six offline optimization tasks, spanning both continuous and discrete design spaces. For the continuous domain, we utilize three tasks from Design-Bench (Trabucco et al., 2022). (1) *Superconductor* (SuperC) (Hamidieh, 2018) contains $17,014$ designs with 86 components, where the goal is to

*Table 1.* Experiment results. We report the **Normalized Maximum Score** ($100^{th}$ percentile) among $K = 128$ candidates. Results are averaged over 8 random seeds (mean $\pm$ standard error). **Bold** indicates the best performance. Underline presents the second best method.

| Method | SuperC | Ant | D'Kitty | LLM-DM | TF8 | TF10 | Mean Rank | Median Rank |
|---|---|---|---|---|---|---|---|---|
| $\mathcal{D}$(best) | 0.399 | 0.565 | 0.884 | 1.000 | 0.439 | 0.511 | – | – |
| *Standard Optimization Baselines* | | | | | | | | |
| CMA-ES | $0.465 \pm 0.024$ | $\mathbf{1.561 \pm 0.896}$ | $0.724 \pm 0.001$ | $0.975 \pm 0.016$ | $0.939 \pm 0.039$ | $0.692 \pm 0.013$ | 13.3/24 | 13.5/24 |
| REINFORCE | $0.481 \pm 0.013$ | $0.263 \pm 0.026$ | $0.573 \pm 0.204$ | $0.305 \pm 0.027$ | $0.961 \pm 0.034$ | $0.632 \pm 0.012$ | 19.8/24 | 23.0/24 |
| BO-qEI | $0.402 \pm 0.034$ | $0.812 \pm 0.000$ | $0.896 \pm 0.000$ | $0.953 \pm 0.022$ | $0.825 \pm 0.091$ | $0.663 \pm 0.011$ | 18.7/24 | 19.5/24 |
| *Forward Surrogate Methods* | | | | | | | | |
| Standard GA | $0.505 \pm 0.013$ | $0.293 \pm 0.029$ | $0.860 \pm 0.021$ | $0.998 \pm 0.000$ | $0.923 \pm 0.011$ | $0.732 \pm 0.041$ | 14.0/24 | 14.5/24 |
| GA on GP | $0.499 \pm 0.019$ | $0.948 \pm 0.013$ | $0.946 \pm 0.001$ | $0.846 \pm 0.029$ | $0.770 \pm 0.087$ | $0.599 \pm 0.004$ | 16.0/24 | 17.0/24 |
| MC-Dropout | $0.535 \pm 0.064$ | $0.805 \pm 0.021$ | $0.934 \pm 0.004$ | $0.781 \pm 0.251$ | $0.911 \pm 0.042$ | $0.817 \pm 0.031$ | 13.0/24 | 13.0/24 |
| COMs | $0.481 \pm 0.028$ | $0.878 \pm 0.031$ | $0.929 \pm 0.016$ | $0.815 \pm 0.008$ | $0.937 \pm 0.025$ | $0.755 \pm 0.017$ | 15.3/24 | 15.5/24 |
| RoMA | $0.509 \pm 0.015$ | $0.592 \pm 0.059$ | $0.825 \pm 0.016$ | $0.968 \pm 0.011$ | $0.662 \pm 0.000$ | $0.801 \pm 0.000$ | 16.2/24 | 16.0/24 |
| ICT | $0.503 \pm 0.017$ | $0.911 \pm 0.030$ | $0.945 \pm 0.011$ | $0.839 \pm 0.021$ | $0.888 \pm 0.047$ | $0.814 \pm 0.027$ | 14.0/24 | 13.5/24 |
| Tri-mentoring | $0.514 \pm 0.018$ | $0.944 \pm 0.033$ | $0.950 \pm 0.015$ | $0.750 \pm 0.002$ | $0.899 \pm 0.045$ | $0.811 \pm 0.039$ | 12.2/24 | 9.5/24 |
| BDI | $0.513 \pm 0.000$ | $0.964 \pm 0.000$ | $0.941 \pm 0.000$ | $0.988 \pm 0.025$ | $0.973 \pm 0.000$ | $\underline{0.882 \pm 0.000}$ | 5.7/24 | 4.0/24 |
| LTR | $0.514 \pm 0.022$ | $0.904 \pm 0.036$ | $0.958 \pm 0.012$ | $0.982 \pm 0.041$ | $0.973 \pm 0.010$ | $0.849 \pm 0.023$ | 6.2/24 | 5.0/24 |
| MATCH-OPT | $0.504 \pm 0.021$ | $0.931 \pm 0.011$ | $0.957 \pm 0.014$ | $0.850 \pm 0.033$ | $0.977 \pm 0.004$ | $0.824 \pm 0.008$ | 9.2/24 | 8.5/24 |
| PGS | $\mathbf{0.563 \pm 0.058}$ | $0.949 \pm 0.017$ | $0.966 \pm 0.013$ | $0.595 \pm 0.027$ | $\underline{0.981 \pm 0.015}$ | $0.793 \pm 0.021$ | 8.0/24 | 4.0/24 |
| *Inverse Generative Methods* | | | | | | | | |
| CbAS | $0.503 \pm 0.069$ | $0.846 \pm 0.033$ | $0.895 \pm 0.016$ | $0.919 \pm 0.043$ | $0.903 \pm 0.028$ | $0.652 \pm 0.032$ | 16.2/24 | 16.0/24 |
| MINs | $0.499 \pm 0.017$ | $0.894 \pm 0.022$ | $0.939 \pm 0.004$ | $0.982 \pm 0.007$ | $0.908 \pm 0.063$ | $0.647 \pm 0.012$ | 14.0/24 | 14.0/24 |
| DDOM | $0.481 \pm 0.015$ | $0.926 \pm 0.025$ | $0.923 \pm 0.009$ | $0.983 \pm 0.028$ | $0.884 \pm 0.042$ | $0.708 \pm 0.015$ | 14.7/24 | 17.0/24 |
| GABO | $0.508 \pm 0.007$ | $0.224 \pm 0.051$ | $0.719 \pm 0.001$ | $0.975 \pm 0.019$ | $0.939 \pm 0.038$ | $0.739 \pm 0.009$ | 15.0/24 | 13.5/24 |
| GTG | $0.480 \pm 0.055$ | $0.865 \pm 0.040$ | $0.935 \pm 0.010$ | $0.910 \pm 0.030$ | $0.901 \pm 0.039$ | $0.801 \pm 0.004$ | 15.0/24 | 14.0/24 |
| RGD | $0.515 \pm 0.011$ | $0.922 \pm 0.020$ | $0.883 \pm 0.014$ | $\underline{1.004 \pm 0.008}$ | $0.889 \pm 0.068$ | $0.825 \pm 0.039$ | 10.7/24 | 9.5/24 |
| BONET | $0.434 \pm 0.021$ | $0.948 \pm 0.025$ | $0.955 \pm 0.010$ | $0.874 \pm 0.039$ | $0.894 \pm 0.086$ | $0.796 \pm 0.018$ | 13.3/24 | 14.5/24 |
| DEMO | $0.520 \pm 0.012$ | $0.948 \pm 0.013$ | $0.954 \pm 0.013$ | $0.907 \pm 0.005$ | $0.808 \pm 0.044$ | $0.836 \pm 0.042$ | 9.7/24 | 6.5/24 |
| ROOT | $0.525 \pm 0.012$ | $0.965 \pm 0.014$ | $\underline{0.972 \pm 0.005}$ | $0.905 \pm 0.006$ | $\mathbf{0.986 \pm 0.007}$ | $0.833 \pm 0.046$ | 5.0/24 | 3.5/24 |
| **SPADE (Ours)** | $\underline{0.546 \pm 0.013}$ | $\underline{0.978 \pm 0.006}$ | $\mathbf{0.981 \pm 0.003}$ | $\mathbf{1.019 \pm 0.064}$ | $0.923 \pm 0.015$ | $\mathbf{0.915 \pm 0.010}$ | **2.8/24** | **1.5/24** |

synthesize a superconducting material with a maximized critical temperature. (2) *Ant Morphology* (Ant) (Brockman et al., 2016) and (3) *D'Kitty Morphology* (D'Kitty) (Ahn et al., 2020) focus on robot morphology optimization, consisting of 10,004 designs with 60 and 56 dimensions, respectively. The objective for Ant is to design a quadruped robot that crawls efficiently, while D'Kitty aims to create a robot capable of navigating to a target location. Additionally, we integrate (4) *LLM Data Mixture* (LLM-DM) (Yen et al., 2025), a high-dimensional continuous task (472 designs, 5 dimensions) where the objective is to optimize pre-training data weights across five domains (Wikipedia, Books, Github, StackExchange, and ArXiv) to maximize downstream model performance. For discrete optimization, we select the (5) *TF Bind 8* (TF8) (Barrera et al., 2016) and (6) *TF Bind 10* (TF10) (Le et al., 2018) tasks, which aim to discover high-affinity DNA sequences of lengths 8 and 10, utilizing datasets of 32,898 and 50,000 samples, respectively. Note that for TF10, the original labels represent binding free energy differences (ddG), where lower values indicate stronger binding; consistent with the task description in Design-Bench, we negate these values so that higher scores correspond to stronger binding affinity.

**Baselines.** We compare SPADE against a comprehensive set of 23 baseline methods covering the full spectrum of

offline BBO paradigms. **(i) Standard Optimization:** (1) *CMA-ES* (Hansen, 2006) iteratively adapts a multivariate normal distribution via covariance matrix updates to converge toward optimal solutions. (2) *REINFORCE* (Williams, 1992) treats design generation as a reinforcement learning problem. (3) *BO-qEI* (Wilson et al., 2017) executes Bayesian Optimization using a Gaussian Process surrogate to value candidates via the q-Expected Improvement acquisition function. **(ii) Forward Surrogate-Based Approaches:** (4) *Standard GA* searches for optimal designs using gradient ascent applied to a deterministic neural network surrogate. (5) *GA on GP* employs gradient ascent to maximize the predictive mean of a Gaussian Process surrogate. (6) *MC-Dropout* (Gal & Ghahramani, 2016) utilizes dropout during inference to approximate epistemic uncertainty, guiding risk-aware optimization. (7) *COMs* (Trabucco et al., 2021) regularizes the surrogate by penalizing the predicted scores of adversarial candidates generated via gradient ascent to mitigate overestimation. (8) *RoMA* (Yu et al., 2021) enhances the surrogate by enforcing local smoothness regularization. (9) *ICT* (Yuan et al., 2023) mitigates distribution shifts by leveraging pseudo-labeling to incorporate useful information from synthetic candidates. (10) *Tri-mentoring* (Chen et al., 2023) utilizes a tri-mentoring mechanism where an ensemble of surrogates improves robustness by filtering and exchanging pseudo-labels based on pairwise comparisons.

(11) *BDI* (Chen et al., 2022) enforces consistency between a forward performance predictor and a backward design generator to effectively transfer knowledge from the offline dataset to new designs. (12) *LTR* (Tan et al., 2025) replaces the traditional regression objective with a ranking-based loss, prioritizing the correct relative ordering of designs over precise value prediction. (13) *MATCH-OPT* (Hoang et al., 2024) aligns the surrogate's gradient field with the latent gradient field of the data support to minimize the distribution shift of generated candidates. (14) *PGS* (Chemingui et al., 2024) learns an optimization policy guided by a surrogate that incorporates perturbation-based smoothing to improve local exploration. **(iii) Inverse Generative Modeling:** (15) *CbAS* (Brookes et al., 2019) trains a generative model re-weighted by the probability that samples satisfy a desired high-performance condition to strictly stay within the data manifold. (16) *MINs* (Kumar & Levine, 2020) learns an explicit inverse mapping from performance scores to designs using a conditional GAN, allowing direct sampling of high-scoring candidates. (17) *DDOM* (Krishnamoorthy et al., 2023) models the inverse conditional distribution $p(\boldsymbol{x}|y)$ using a diffusion model to generate candidates conditioned on high target values. (18) *GABO* (Yao et al., 2024) explores the latent space of a GAN using Bayesian Optimization, constrained by an adaptive source critic regularization to ensure reliability. (19) *GTG* (Yun et al., 2024) trains a conditional diffusion model on synthetic trajectories constructed to lead toward high-scoring regions, utilizing guidance to extrapolate beyond the dataset. (20) *RGD* (Chen et al., 2024) enhances a proxy-free diffusion model with explicit guidance from a surrogate proxy, which is iteratively refined using diffusion-generated samples. (21) *BONET* (Mashkaria et al., 2023) generates designs using an autoregressive model that focuses on high-value regions identified by a separate regressor. (22) *DEMO* (Yuan et al., 2025b) explicitly trains a diffusion model to capture the design distribution $p(\boldsymbol{x})$. (23) *ROOT* (Dao et al., 2025) formulates optimization as a distributional translation problem, learning a probabilistic bridge to transform the distribution of low-scoring data into high-scoring candidates.

**Evaluation Metrics.** Following standard protocols (Trabucco et al., 2022), we report the *Normalized Maximum Score* (100th percentile) achieved by the top candidate suggested by each method, given a query budget of 128. For a discovered design with a raw score $y$, the normalized score is calculated as $(y - y_{\min})/(y_{\max} - y_{\min})$, where $y_{\min}$ and $y_{\max}$ represent the minimum and maximum scores available in the entire dataset (including those unseen during training). All results are reported as the mean and standard error across 8 independent random seeds. Additional results for the *Normalized Median Score* (50th percentile) are provided in Appendix E. We also report the normalized property score of the best design in the static dataset $\mathcal{D}$ as

$\mathcal{D}$(best) for reference. We directly cite baseline results from Dao et al. (2025) when available. Otherwise, we run the baselines following their original settings for 8 trials and report the mean and standard error.

**Implementation Details.** The score network $\epsilon_\theta(\cdot)$ in SPADE is a 3-layer MLP with 2048 hidden units and SiLU activations. We train the diffusion model for 100 epochs using the Adam optimizer (Kingma & Ba, 2015) with a learning rate of $10^{-3}$, a batch size of 64, and a linear variance schedule ranging from $\beta_{\text{start}} = 10^{-4}$ to $\beta_{\text{end}} = 2 \times 10^{-2}$ over 100 diffusion steps. We convert discrete designs to continuous logits (Trabucco et al., 2022). The regularization weights $\lambda_1$ and $\lambda_2$ are detailed in Appendix F. For the rank loss, we construct 32 pairs per batch with a temperature of $s = 1.0$. Support proximity is calculated using $k = 10$ nearest neighbors, with penalty parameters fixed at $a = 0.02$, $a_0 = 0.02$, and $a_1 = 0.005$. During training, we estimate statistical moments using 8 short-run Monte Carlo samples generated via 10 DDIM (Song et al., 2021) steps. For optimization, we employ a Genetic Algorithm with a population size of 128 and 64 elites for 100 generations. The Genetic Algorithm maximizes the LCB acquisition function ($\beta = 1.0$) computed from 256 Monte Carlo samples per candidate. The mutation noise is initialized at $\sigma = 0.12$ and decays to 0.02. All experiments were conducted on a workstation equipped with a single NVIDIA A100 GPU.

### 4.2. Main Results and Analysis

We present the quantitative results of our comparative study in Table 1, answering **RQ1**. Overall, SPADE achieves state-of-the-art performance, securing the best **Mean Rank of 2.8** and **Median Rank of 1.5** among all 24 evaluated methods. Notably, it achieves the top 2 normalized maximum scores on **5 out of 6 tasks** (*SuperC*, *Ant*, *D'Kitty*, *LLM-DM*, and *TF10*). This consistent superiority across diverse domains validates SPADE's ability to reliably identify high-performing designs in out-of-distribution regions without succumbing to the instability observed in other baselines.

**Performance on Continuous Tasks.** SPADE demonstrates exceptional extrapolation capabilities in high-dimensional continuous search spaces. On the robot morphology tasks, SPADE outperforms all robust offline baselines. Specifically, on *Ant Morphology*, SPADE achieves a score of $0.978 \pm 0.006$. It is worth noting that while CMA-ES achieves a higher mean score, it exhibits an extremely high standard error, indicating severe instability. This suggests its performance is driven by stochastic outliers rather than reliable optimization. In contrast, SPADE delivers consistently high performance with low variance, significantly surpassing contenders like ROOT. On *D'Kitty Morphology*, SPADE obtains the highest score of 0.981, edging out the

*Table 2.* Ablation study on all tasks. We report Normalized Maximum Score among $K{=}128$ candidates. Results are mean $\pm$ standard error over 8 seeds.

| Task | Base | w/o Prox | w/o Calib | Full |
|------|------|----------|-----------|------|
| SuperC | $0.519 \pm 0.012$ | $0.538 \pm 0.011$ | $0.542 \pm 0.010$ | $\mathbf{0.546 \pm 0.013}$ |
| Ant | $0.932 \pm 0.011$ | $0.952 \pm 0.008$ | $0.963 \pm 0.007$ | $\mathbf{0.978 \pm 0.006}$ |
| D'Kitty | $0.962 \pm 0.006$ | $0.972 \pm 0.005$ | $0.975 \pm 0.004$ | $\mathbf{0.981 \pm 0.003}$ |
| LLM-DM | $0.957 \pm 0.070$ | $0.979 \pm 0.060$ | $0.998 \pm 0.058$ | $\mathbf{1.019 \pm 0.064}$ |
| TF8 | $0.890 \pm 0.018$ | $0.912 \pm 0.016$ | $0.897 \pm 0.017$ | $\mathbf{0.923 \pm 0.015}$ |
| TF10 | $0.870 \pm 0.014$ | $0.895 \pm 0.012$ | $0.882 \pm 0.012$ | $\mathbf{0.915 \pm 0.010}$ |

recent state-of-the-art (SOTA) method ROOT, while CMA-ES lags significantly behind. These results suggest that our support-proximity regularization effectively penalizes the over-optimistic "hallucinations" that often plague standard optimization methods on learned surrogates. Furthermore, on the *LLM Data Mixture* task, SPADE achieves a remarkable score of $1.019 \pm 0.064$, effectively discovering data weighting schemes that outperform the best configurations in the offline dataset. It outperforms other strong generative baselines like RGD and diffusion-based DDOM, highlighting its efficacy in LLM pre-training applications. On *Superconductor*, SPADE remains highly competitive, surpassing ROOT, though it slightly trails PGS.

**Performance on Discrete Tasks.** In the discrete domain, SPADE exhibits strong robustness, particularly on harder tasks. On *TF Bind 10*, which involves a larger search space than TF Bind 8, SPADE achieves a dominant score of $0.915 \pm 0.010$, outperforming the runner-up BDI by a clear margin. This result is particularly notable given that many generative baselines struggle here (e.g., MINs achieves 0.647, CbAS achieves 0.652) and standard optimizers like CMA-ES perform poorly. This indicates that our forward diffusion modeling approach avoids the mode collapse often seen in inverse methods and scales better to complex discrete landscapes than standard ensemble methods. On *TF Bind 8*, SPADE remains competitive with a score of 0.923, though it is outperformed by some recent SOTA methods. We hypothesize that for shorter sequences with simpler landscapes, distribution translation (ROOT) or distribution matching (MATCH-OPT) may offer slight advantages, whereas SPADE's probabilistic conservatism shines more in complex scenarios like TF10.

**Summary.** In conclusion, SPADE establishes a new state-of-the-art benchmark in offline BBO. By effectively balancing the exploration of high-scoring regions with rigorous support-proximity constraints, it delivers robust and superior performance across a wide spectrum of tasks.

### 4.3. Ablation Studies

To address **RQ2** regarding the individual contributions of SPADE's components, we evaluate three variants of our framework. (1) **Base** trains the diffusion surrogate using only the standard denoising loss without additional regularization. (2) **w/o Prox** incorporates the calibrated diffusion estimation objectives but excludes the support-proximity regularization. (3) **w/o Calib** applies the proximity regularization while relying on standard diffusion training objectives, omitting the calibration loss. **Full** represents the complete SPADE framework integrating both modules.

Table 2 summarizes the results across all benchmark tasks. The **Base** model consistently yields the lowest performance, indicating that standard diffusion training is insufficient for effective optimization. Integrating either the calibration module (**w/o Prox**) or the proximity regularization (**w/o Calib**) provides distinct performance improvements over the baseline. Most importantly, the **Full** method achieves the best scores on all tasks, demonstrating that the calibration and proximity modules are complementary and both essential for the framework's success.

To isolate the contribution of the diffusion backbone, we conduct an additional experiment comparing SPADE against a Gaussian Process (GP) surrogate equipped with the same calibration loss and support-proximity regularization, and evolutionary algorithm search pipeline. Results are provided in Appendix G.

## 5. Related Work

**Generative Models for Surrogate Modeling.** While traditional surrogates rely on Gaussian Processes (Rasmussen & Williams, 2006) or Deep Ensembles (Lakshminarayanan et al., 2017), recent work leverages deep generative models to capture complex uncertainty. In online settings, methods like Neural Processes (Garnelo et al., 2018) and LLM-based regressors such as OmniPred (Song et al., 2024) have demonstrated strong distribution modeling capabilities. In offline optimization, generative models often serve auxiliary roles: Auto-Focus (Fannjiang & Listgarten, 2020) uses VAEs for density-based reweighting, while RGD (Chen et al., 2024) employs diffusion to refine proxy predictions. Distinct from these approaches, SPADE explicitly parameterizes the forward surrogate $p(y|\boldsymbol{x})$ using a conditional diffusion model, directly leveraging its generative structure to derive calibrated aleatoric and epistemic uncertainty for acquisition-based optimization. Some diffusion-based reward optimization methods use a feasibility-check reward that keeps the inference close to valid designs. SEIKO (Uehara et al., 2024) operates in an online setting where the agent can iteratively query the ground-truth reward oracle. In contrast, SPADE operates in the offline setting where no oracle access is available.

**Offline Model-Based Optimization.** Existing approaches generally fall into two paradigms: forward methods that optimize designs via gradient ascent on a regressor (Trabucco

et al., 2021; Yu et al., 2021), and inverse methods that model $p(\boldsymbol{x}|y)$ to sample high-scoring designs directly (Brookes et al., 2019; Kumar & Levine, 2020). Since we have formalized these frameworks in Section 2 and detailed 23 representative baselines in Section 4.1, we omit a redundant review here. More recent works like dLLM (Yuan et al., 2026) and DiBO (Sun et al., 2026) employ diffusion language models (Nie et al., 2026) to search for optimal candidates. SPADE unifies these paradigms by employing a forward optimization strategy guided by a diffusion-based probabilistic surrogate, effectively balancing high-fidelity likelihood modeling with robust geometric constraints.

## 6. Conclusion

In this paper, we introduced SPADE, a support-proximity augmented diffusion surrogate framework for offline black-box optimization. SPADE explicitly models the forward likelihood $p(y \mid \boldsymbol{x})$ using a conditional diffusion model, integrated with (i) calibration losses to ensure moment and ranking fidelity, and (ii) a kNN-based support regularizer that enforces conservatism in low-density regions. Extensive evaluations across the Design-Bench suite and a high-dimensional LLM data mixture optimization benchmark demonstrate that SPADE consistently outperforms strong baselines and establishes state-of-the-art performance. Future work will focus on scaling support estimation to higher dimensions through learnable density proxies and extending this probabilistic paradigm to multi-objective optimizations.

## Impact Statement

This paper presents work whose goal is to advance the field of offline model-based optimization. Our proposed framework, SPADE, aims to accelerate scientific discovery and engineering design in domains ranging from material science and robotics to biological sequence design and LLM pre-training. A core motivation of our approach is to mitigate "reward hacking" and improve the reliability of optimization in out-of-distribution regions. By enforcing support-proximity constraints, SPADE aims to produce designs that are not only high-performing but also physically or biologically plausible, thereby contributing to safer and more robust optimization systems. However, as with many generative design methods, there is a potential for dual-use, particularly in sensitive fields like biological sequence design. We believe that developing methods with rigorous uncertainty quantification and adherence to data support, as advocated in this work, is a crucial step toward mitigating such risks and ensuring responsible deployment.

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

# A. Proof of Proposition 3.1

**Proposition A.1** (First-order equivalence of support-aware acquisition and prior penalization). *Let $\mathcal{A}(\mu, \sigma)$ be a continuously differentiable acquisition function that is strictly increasing in $\mu$ and strictly decreasing in $\sigma$, i.e., $\partial_\mu \mathcal{A}(\mu, \sigma) > 0$ and $\partial_\sigma \mathcal{A}(\mu, \sigma) < 0$. For a design $x$, let $\mu = \hat{\mu}_\theta(x)$ and $\sigma = \hat{\sigma}_\theta(x)$, and define the support distance $d(x) = \log R_k(x)$. Consider the support transform*

$$T_d : (\mu, \sigma) \mapsto \big(\mu - \tau(d),\ \max\{\sigma,\ \sigma_{\min}(d)\}\big), \quad \tau(d) = a\, d,\ \ \sigma_{\min}(d) = a_0 + a_1 d,$$

*with coefficients $a, a_0, a_1 \geq 0$. Define the support-aware acquisition $\widetilde{\mathcal{A}}(x) = \mathcal{A}\big(T_{d(x)}(\mu, \sigma)\big)$.*

*Then there exist a term $C(x) \in \mathbb{R}$ and a strictly positive coefficient $\kappa(x)$ given by*

$$\kappa(x) = \frac{1}{D}\Big(a\, \partial_\mu \mathcal{A}(\mu, \sigma)\ -\ a_1\, \partial_\sigma \mathcal{A}(\mu, \sigma)\, \mathbb{I}\{\sigma < a_0\}\Big)\ >\ 0$$

*(where $D$ is the ambient dimension, i.e., the dimension of designs), such that for small regularization updates (i.e., small $a, a_1$ or small $d(x)$), the following expansion holds:*

$$\widetilde{\mathcal{A}}(x) = \mathcal{A}(\mu, \sigma) - \kappa(x)\big(-\log \hat{p}_{knn}(x)\big) + C(x) + o\big(\tau(d)\big). \tag{13}$$

*Consequently, maximizing the support-aware acquisition $\widetilde{\mathcal{A}}(x)$ is, to first order, equivalent to maximizing the joint objective of utility and prior probability:*

$$\underbrace{\mathcal{A}(\mu, \sigma)}_{\text{Likelihood Utility}} + \underbrace{\kappa(x)\, \log \hat{p}_{knn}(x)}_{\text{Prior Constraint}}.$$

*Proof.* Fix a design $x$ and abbreviate $d = d(x)$, $\mu = \hat{\mu}_\theta(x)$, and $\sigma = \hat{\sigma}_\theta(x)$. Let the perturbations be $\delta\mu = \tau(d) = ad$ and

$$\delta\sigma(d; \sigma) = \max\{0,\ \sigma_{\min}(d) - \sigma\} = \max\{0,\ a_0 - \sigma + a_1 d\}.$$

By performing a first-order Taylor expansion of $\mathcal{A}$ around $(\mu, \sigma)$, we obtain:

$$\mathcal{A}(\mu - \delta\mu,\ \sigma + \delta\sigma) = \mathcal{A}(\mu, \sigma) - \partial_\mu \mathcal{A}(\mu, \sigma)\, \delta\mu + \partial_\sigma \mathcal{A}(\mu, \sigma)\, \delta\sigma + r(\delta\mu, \delta\sigma), \tag{14}$$

where $r$ represents higher-order terms that vanish as the perturbations approach zero. We analyze the variance perturbation $\delta\sigma$ in two cases.

**Case 1:** $\sigma \geq a_0$ **(Variance is sufficient).** For sufficiently small regularization scales (small $a_1$) or small distance $d$, we have $a_1 d \leq \sigma - a_0$. Thus, the variance floor is not triggered, implying $\delta\sigma = 0$. Substituting $\delta\mu = ad$ and $\delta\sigma = 0$ into Eq. (14) yields:

$$\widetilde{\mathcal{A}}(x) = \mathcal{A}(\mu, \sigma) - \partial_\mu \mathcal{A}(\mu, \sigma)\, a\, d + \text{h.o.t.}$$

**Case 2:** $\sigma < a_0$ **(Variance is insufficient).** Here, the variance floor is active. Assuming small perturbations, $\delta\sigma = (a_0 - \sigma) + a_1 d$. Inserting this into Eq. (14) gives:

$$\widetilde{\mathcal{A}}(x) = \mathcal{A}(\mu, \sigma) - \partial_\mu \mathcal{A}(\mu, \sigma)\, a\, d + \partial_\sigma \mathcal{A}(\mu, \sigma)\, (a_0 - \sigma) + \partial_\sigma \mathcal{A}(\mu, \sigma)\, a_1 d + \text{h.o.t.}$$

The term $\partial_\sigma \mathcal{A}(\mu, \sigma)\, (a_0 - \sigma)$ depends only on the current state $x$ (via $\sigma$) and not on the distance $d$; thus, it can be absorbed into a state-dependent constant $C(x)$. Collecting the terms involving $d$ yields:

$$\widetilde{\mathcal{A}}(x) = \mathcal{A}(\mu, \sigma) - \Big(a\, \partial_\mu \mathcal{A}(\mu, \sigma) - a_1\, \partial_\sigma \mathcal{A}(\mu, \sigma)\Big) d + C(x) + \text{h.o.t.}$$

**Combining and Linking to Prior.** Combining both cases using the indicator function $\mathbb{I}\{\cdot\}$, we have:

$$\widetilde{\mathcal{A}}(x) \approx \mathcal{A}(\mu, \sigma) - \Big(a\, \partial_\mu \mathcal{A}(\mu, \sigma) - a_1\, \partial_\sigma \mathcal{A}(\mu, \sigma)\, \mathbb{I}\{\sigma < a_0\}\Big) d + C(x). \tag{15}$$

Recalling the definition of the kNN density estimator from Eq. (3):

$$\hat{p}_{\text{knn}}(\boldsymbol{x}) = \frac{k}{NV_D R_k(\boldsymbol{x})^D}.$$

Taking the negative logarithm reveals the linear relationship between log-density and the distance metric $d = \log R_k(\boldsymbol{x})$:

$$-\log \hat{p}_{\text{knn}}(\boldsymbol{x}) = \underbrace{- \log \frac{k}{NV_D}}_{=:C_0} + D \log R_k(\boldsymbol{x}) = C_0 + D\, d.$$

Substituting $d = \frac{1}{D}\left(-\log \hat{p}_{\text{knn}}(\boldsymbol{x}) - C_0\right)$ into Eq. (15), we arrive at:

$$\widetilde{\mathcal{A}}(\boldsymbol{x}) = \mathcal{A}(\mu, \sigma) - \underbrace{\frac{a\, \partial_\mu \mathcal{A}(\mu, \sigma) - a_1\, \partial_\sigma \mathcal{A}(\mu, \sigma)\, \mathbb{I}\{\sigma < a_0\}}{D}}_{\kappa(\boldsymbol{x})} \left(- \log \hat{p}_{\text{knn}}(\boldsymbol{x})\right) + C'(\boldsymbol{x}).$$

Since $\mathcal{A}$ is increasing in $\mu$ ($\partial_\mu \mathcal{A} > 0$) and decreasing in $\sigma$ ($\partial_\sigma \mathcal{A} < 0$), and all coefficients $a, a_1, D$ are non-negative, it follows that $\kappa(\boldsymbol{x}) > 0$. This completes the proof that the support-aware transform adds a positive log-prior reward (equivalent to subtracting a negative log-likelihood penalty) to the acquisition function. $\square$

## B. Additional Experiment with Other Acquisition Functions

Our framework is fundamentally compatible with standard acquisition functions beyond the default Lower Confidence Bound (LCB). Alternative strategies such as Expected Improvement (EI) and Mean-Variance Regression (MVR) can be seamlessly integrated by substituting $\mathcal{A}(\mu, \sigma)$ in Section 2.2. In all cases, the proposed support-aware transformation operates consistently by decreasing the effective predictive mean and enforcing a minimum uncertainty floor as a function of the support distance. Consequently, the theoretical proposition discussed in the main text applies to any acquisition function that is monotonically increasing in the predicted mean $\mu$ and decreasing in the uncertainty $\sigma$.

To empirically validate this robustness, we conducted additional evaluations on two representative tasks: *Ant Morphology* (Continuous) and *TF Bind 10* (Discrete). We compared the performance of SPADE using LCB against variants using EI and MVR. Our experimental results indicate that varying the acquisition function does not yield statistically significant differences in the final normalized maximum scores, with relative performance fluctuations remaining minimal (approximately 3%). This consistency suggests that SPADE's superior performance is primarily driven by the high-quality, calibrated uncertainty estimates and the rigorous support constraints provided by the underlying diffusion surrogate, rather than the specific heuristic used to aggregate these estimates into a scalar acquisition value.

## C. Runtime Breakdown

We provide a detailed breakdown of the computational cost of SPADE in Table 3. All experiments were conducted on a single NVIDIA A100 (80GB) GPU. The total runtime consists of two phases: (1) **Training**, which involves training the surrogate with the dataset $\mathcal{D}$; and (2) **Optimization**, which involves running the genetic algorithm using the surrogate-derived acquisition function to propose 128 candidates.

As shown in Table 3, the computational cost is highly efficient. For the high-dimensional *TF Bind 10* task, which features the largest dataset ($50,000$ samples), training takes approximately $43$ minutes, while the optimization phase requires only $8$ minutes. For continuous control tasks like *Ant* and *D'Kitty*, the entire pipeline completes in under $26$ minutes. Even in the most computationally demanding scenario, the total wall-clock time remains under one hour. This efficiency demonstrates that SPADE does not incur prohibitive computational overhead, making it a practical solution for real-world offline optimization problems where training time is often negligible compared to the cost of online wet-lab experiments.

## D. Qualitative Synthetic Surfaces

To complement our quantitative results on high-dimensional benchmarks, we provide a qualitative analysis of the optimization landscapes learned by SPADE. Visualizing the decision boundaries and potential energy landscapes in high-dimensional

*Table 3.* **Wall-clock time breakdown** on a single NVIDIA A100 GPU. We report the duration for training the diffusion surrogate and performing the offline candidate search (generating 128 designs). All times are in *minutes:seconds*.

| Task | Training | Optimization | Total |
|---|---|---|---|
| SuperC | 14:39 | 08:23 | 23:02 |
| Ant | 13:06 | 12:46 | 25:52 |
| D'Kitty | 13:06 | 12:46 | 25:52 |
| TF8 | 27:37 | 08:11 | 35:48 |
| TF10 | 43:01 | 08:23 | 51:24 |
| LLM-DM | 03:08 | 03:01 | 06:09 |

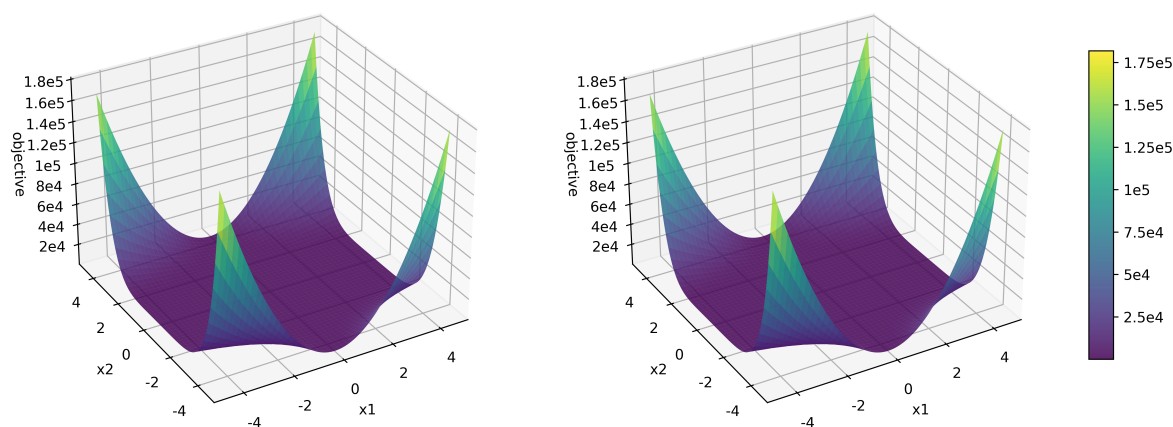

*Figure 2.* **Beale (2D):** ground-truth objective surface (left) and the landscape learned by SPADE (right).

spaces (e.g., 60D for Ant or 86D for SuperC) is inherently difficult. Therefore, we employ standard 2D synthetic test functions from the BayesO benchmark (Kim, 2023) to directly inspect the fidelity of our surrogate model.

We selected two representative functions: the **Beale** function (Figure 2), characterized by its sharp peaks and complex multi-modality, and the **Zakharov** function (Figure 3), known for its smooth, plate-shaped valley. We trained SPADE on offline datasets sampled from these functions and queried the diffusion surrogate to reconstruct the landscape.

As illustrated in the figures, the landscape learned by SPADE (Right) exhibits a remarkable resemblance to the ground-truth objective function (Left). Crucially, SPADE accurately captures the global geometry, the curvature of the gradients, and the precise locations of the extrema without introducing significant artifacts. This qualitative evidence reinforces our claim that the proposed SPADE effectively models the underlying data distribution, enabling reliable optimization.

# E. Normalized Median Score Results

We report the normalized median scores (50[th] percentile) of the candidates generated by each method in Table 4. Since median scores are not universally reported in Dao et al. (2025), we re-evaluated all baselines using their official implementations under identical experimental settings to ensure a fair comparison.

Consistent with the maximum score results in the main text, SPADE demonstrates exceptional robustness, achieving the highest median scores on **4 out of 6 tasks** (*Ant*, *LLM-DM*, *TF8*, and *TF10*) and ranking second on *SuperC*. Consequently, SPADE secures the best **Mean Rank of 1.7** and **Median Rank of 1.0** among all 24 methods. This dominance in median scores indicates that the combination of calibrated diffusion modeling and support-proximity regularization effectively shifts the *entire distribution* of generated candidates toward high-performance regions, avoiding the mode collapse issues often observed in inverse generative baselines (e.g., CbAS, GABO)

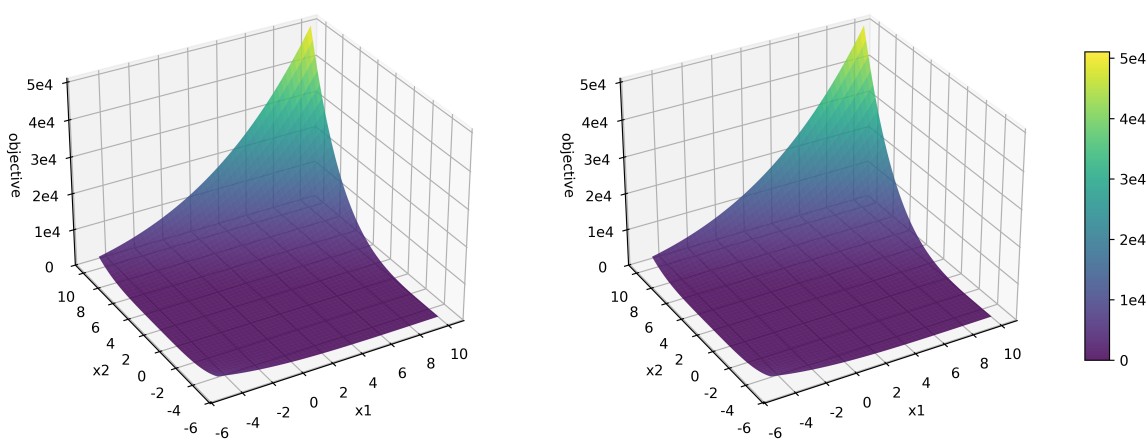

*Figure 3.* **Zakharov (2D):** ground-truth objective surface (left) and the landscape learned by SPADE (right).

## F. Additional Hyperparameters

As mentioned in Section 4.1, we employ the **Optuna** framework (Akiba et al., 2019) to automatically tune the two key regularization hyperparameters for each task: the calibration weight $\lambda_1$ and the support-proximity weight $\lambda_2$. To isolate the impact of these regularization terms, all other hyperparameters are kept fixed across experiments.

For the majority of tasks, including *Ant Morphology*, *D'Kitty Morphology*, *LLM Data Mixture*, and the discrete *TF Bind* tasks, we sample the calibration weight $\lambda_1$ from a uniform distribution in the range $[0.0, 0.8]$ and the proximity weight $\lambda_2$ from a log-uniform distribution in the range $[0.2, 12.0]$. For the *Superconductor* task, we utilize a slightly constrained search space, sampling $\lambda_1$ from $\mathcal{U}[0.0, 0.6]$ and $\lambda_2$ from $\log \mathcal{U}[0.6, 12.0]$.

## G. Ablation Study with Gaussian Process Backbone

In this section, we provide additional ablation study results in Table 5 by replacing the diffusion backbone to a standard Gaussian Process (GP) with RBF kernel. When adding the calibration loss (GP + Calib) and the support-proximity regularization (GP + Prox) separately, both ablated variants improve the vanilla GP backbone with the same evolutionary algorithm (EA) and LCB acquisition function (GP base same EA + LCB pipeline). Including both calibration loss and support-proximity regularization can consistently enhance the performance further. However, comparing to our proposed method SPADE, which leverages diffusion backbone, GP + Calib + Prox still underperforms.

*Table 4.* Experiment results. We report the **Normalized Median Score** (50$^{\text{th}}$ percentile) among $K = 128$ candidates. Results are averaged over 8 random seeds (mean $\pm$ standard error). **Bold** indicates the best performance. Underline presents the second best method.

| Method | SuperC | Ant | D'Kitty | LLM-DM | TF8 | TF10 | Mean Rank | Median Rank |
|---|---|---|---|---|---|---|---|---|
| $\mathcal{D}$(best) | 0.399 | 0.565 | 0.884 | 1.000 | 0.439 | 0.511 | – | – |
| *Standard Optimization Baselines* | | | | | | | | |
| CMA-ES | $0.379 \pm 0.003$ | $-0.045 \pm 0.004$ | $0.684 \pm 0.016$ | $0.812 \pm 0.028$ | $0.537 \pm 0.014$ | $0.484 \pm 0.014$ | 16.8/24 | 15.5/24 |
| REINFORCE | $\mathbf{0.463 \pm 0.016}$ | $0.138 \pm 0.032$ | $0.356 \pm 0.131$ | $0.205 \pm 0.035$ | $0.462 \pm 0.021$ | $0.475 \pm 0.008$ | 17.2/24 | 19.5/24 |
| BO-qEI | $0.300 \pm 0.015$ | $0.567 \pm 0.000$ | $0.883 \pm 0.000$ | $0.892 \pm 0.019$ | $0.439 \pm 0.000$ | $0.467 \pm 0.000$ | 15.5/24 | 16.0/24 |
| *Forward Surrogate Methods* | | | | | | | | |
| Standard GA | $0.334 \pm 0.004$ | $0.206 \pm 0.041$ | $0.801 \pm 0.033$ | $0.905 \pm 0.017$ | $0.539 \pm 0.030$ | $0.539 \pm 0.010$ | 13.5/24 | 15.5/24 |
| GA on GP | $0.364 \pm 0.030$ | $0.569 \pm 0.021$ | $0.873 \pm 0.009$ | $0.834 \pm 0.040$ | $0.569 \pm 0.050$ | $0.485 \pm 0.021$ | 12.8/24 | 13.5/24 |
| MC-Dropout | $0.395 \pm 0.027$ | $0.521 \pm 0.047$ | $0.876 \pm 0.005$ | $0.776 \pm 0.262$ | $0.534 \pm 0.022$ | $0.514 \pm 0.023$ | 13.3/24 | 13.0/24 |
| COMs | $0.316 \pm 0.024$ | $0.564 \pm 0.002$ | $0.881 \pm 0.002$ | $0.801 \pm 0.021$ | $0.439 \pm 0.000$ | $0.467 \pm 0.002$ | 17.5/24 | 18.0/24 |
| RoMA | $0.370 \pm 0.019$ | $0.477 \pm 0.038$ | $0.854 \pm 0.007$ | $0.878 \pm 0.026$ | $0.555 \pm 0.020$ | $0.512 \pm 0.020$ | 13.2/24 | 13.0/24 |
| ICT | $0.399 \pm 0.012$ | $0.592 \pm 0.025$ | $0.874 \pm 0.005$ | $0.830 \pm 0.018$ | $0.551 \pm 0.013$ | $\underline{0.541 \pm 0.004}$ | 10.5/24 | 10.0/24 |
| Tri-mentoring | $0.355 \pm 0.003$ | $0.606 \pm 0.007$ | $0.866 \pm 0.001$ | $0.740 \pm 0.022$ | $0.609 \pm 0.021$ | $0.527 \pm 0.008$ | 11.5/24 | 11.5/24 |
| BDI | $0.412 \pm 0.000$ | $0.474 \pm 0.000$ | $0.855 \pm 0.000$ | $0.912 \pm 0.014$ | $0.439 \pm 0.000$ | $0.476 \pm 0.000$ | 12.8/24 | 16.0/24 |
| LTR | $0.420 \pm 0.017$ | $0.568 \pm 0.016$ | $0.885 \pm 0.002$ | $0.901 \pm 0.016$ | $0.566 \pm 0.026$ | $0.477 \pm 0.010$ | 9.0/24 | 8.5/24 |
| MATCH-OPT | $0.405 \pm 0.019$ | $0.601 \pm 0.020$ | $0.889 \pm 0.004$ | $0.850 \pm 0.017$ | $0.608 \pm 0.022$ | $0.497 \pm 0.013$ | 8.2/24 | 7.5/24 |
| PGS | $0.379 \pm 0.016$ | $0.532 \pm 0.016$ | $\mathbf{0.941 \pm 0.008}$ | $0.580 \pm 0.029$ | $0.375 \pm 0.014$ | $0.443 \pm 0.005$ | 16.7/24 | 19.5/24 |
| *Inverse Generative Methods* | | | | | | | | |
| CbAS | $0.111 \pm 0.017$ | $0.384 \pm 0.016$ | $0.753 \pm 0.008$ | $0.864 \pm 0.023$ | $0.428 \pm 0.010$ | $0.463 \pm 0.007$ | 20.2/24 | 21.0/24 |
| MINs | $0.336 \pm 0.016$ | $0.618 \pm 0.040$ | $0.887 \pm 0.004$ | $0.903 \pm 0.019$ | $0.421 \pm 0.015$ | $0.468 \pm 0.006$ | 13.2/24 | 12.5/24 |
| DDOM | $0.346 \pm 0.009$ | $0.615 \pm 0.007$ | $0.861 \pm 0.003$ | $0.886 \pm 0.025$ | $0.401 \pm 0.008$ | $0.464 \pm 0.006$ | 16.0/24 | 18.0/24 |
| GABO | $0.350 \pm 0.030$ | $0.190 \pm 0.004$ | $0.674 \pm 0.005$ | $0.832 \pm 0.027$ | $0.496 \pm 0.011$ | $0.457 \pm 0.008$ | 19.5/24 | 20.0/24 |
| GTG | $0.380 \pm 0.022$ | $0.645 \pm 0.098$ | $0.901 \pm 0.005$ | $0.895 \pm 0.020$ | $0.460 \pm 0.032$ | $0.452 \pm 0.010$ | 11.5/24 | 10.0/24 |
| RGD | $0.415 \pm 0.015$ | $0.588 \pm 0.024$ | $0.882 \pm 0.011$ | $\underline{0.954 \pm 0.018}$ | $0.590 \pm 0.029$ | $0.492 \pm 0.014$ | 7.3/24 | 8.0/24 |
| BONET | $0.369 \pm 0.015$ | $\underline{0.819 \pm 0.032}$ | $0.907 \pm 0.020$ | $0.870 \pm 0.022$ | $0.505 \pm 0.055$ | $0.496 \pm 0.037$ | 9.3/24 | 11.0/24 |
| DEMO | $0.400 \pm 0.007$ | $0.604 \pm 0.005$ | $0.891 \pm 0.002$ | $0.906 \pm 0.016$ | $\underline{0.617 \pm 0.027}$ | $0.522 \pm 0.012$ | $\underline{5.5/24}$ | 5.5/24 |
| ROOT | $0.405 \pm 0.010$ | $0.712 \pm 0.014$ | $\underline{0.919 \pm 0.003}$ | $0.905 \pm 0.006$ | $0.595 \pm 0.026$ | $0.473 \pm 0.004$ | $\overline{6.2/24}$ | $\underline{5.0/24}$ |
| **SPADE (Ours)** | $\underline{0.436 \pm 0.055}$ | $\mathbf{0.935 \pm 0.010}$ | $0.905 \pm 0.006$ | $\mathbf{0.994 \pm 0.030}$ | $\mathbf{0.679 \pm 0.023}$ | $\mathbf{0.748 \pm 0.015}$ | **1.7/24** | **1.0/24** |

*Table 5.* Additional Ablation study on one continuous and one discrete tasks. We report Normalized Maximum Score among $K=128$ candidates. Results are mean $\pm$ standard error over 8 seeds.

| Task | GP base(same EA + LCB pipeline) | GP + Calib | GP + Prox | GP + Calib + Prox | SPADE |
|---|---|---|---|---|---|
| Ant | $0.955 \pm 0.021$ | $0.961 \pm 0.009$ | $0.964 \pm 0.018$ | $0.968 \pm 0.013$ | $\mathbf{0.978 \pm 0.006}$ |
| TF10 | $0.734 \pm 0.018$ | $0.762 \pm 0.016$ | $0.789 \pm 0.024$ | $0.802 \pm 0.015$ | $\mathbf{0.915 \pm 0.010}$ |

