# OpenReview forum: "Support-Proximity Augmented Diffusion Estimation for Offline Black-Box Optimization"
_ICML.cc/2026/Conference — ICML 2026 regular_

### Official Review · Reviewer_PUkY · 2026-03-08

**Soundness:** 2
**Presentation:** 2
**Significance:** 2
**Originality:** 2
**Overall Recommendation:** 4
**Confidence:** 3

**Summary:**

The authors tackle the problem of Offline Black-Box Optimization (BBO), where one aims to maximize a function based only on a static dataset of past evaluations: at optimization time, one only proposes a batch of experiments, and the best value found over the batch is computed, serving as a final performance metric.

Interestingly, the proposed method leverages diffusion models for forward modelling $p(y|x)$ instead of inverse generative models $p(x|y)$, as classically done. The diffusion model then produces a mean function
$\hat{\mu}_{\theta}(x)$
and a variance function $\hat{\sigma}_{\theta}(x)$, which can be combined to yield classical acquisition strategies ike Lower Confidence Bound. Framing the problem as such motivates the introduction of two modules by the authors to fully leverage the potential of diffusion models in optimization tasks: calibrated diffusion estimation, and support-proximity regularization. The first ensures that the mean function produced by the diffusion model matches the data and preserves rank consistency, whereas the second penalizes the model in low-density regions. This is achieved by a mean shrinkage and variance inflation for OOD designs, eventually resulting in lower acquisition values.

The method is then evaluated on a range of diverse experiments from continuous to discrete design spaces, and against a large number of concurrent baselines, demonstrating the overall superiority of SPADE.

**Compliance With Llm Reviewing Policy:**

Affirmed.

**Final Justification:**

The rebuttal addressed the few questions I had, and the authors answers to the other reviewers questions are also satisfactory, I will therefore keep my score as is (4).

**Key Questions For Authors:**

- Regarding Appendix F and additional hyperparameters $\lambda_1$ and $\lambda_2$, Optuna was used to automatically tuned them, can you detail a bit more precisely how this was done? The main goal here is just to ensure that. Similarly, adding a full hyperparameter table in the appendix with their name/function/value would be good.

**Limitations:**

yes

**Strengths And Weaknesses:**

(strengths)
- I believe that the angle proposed by the method, which is to use diffusion models for forward modelling rather than as an inverse approach, is novel in the case of offline black-box optimization.

- The evaluation benchmark involves many methods and test functions, demonstrating that SPADE is the overall best competitor. The ablation study on the different loss components thereafter shows that each loss term has a moderate impact on the performance,

(weaknesses)
- I did not find significant weaknesses to report. A comparison of Table 2 with the baseline DEMO (table 1), which uses diffusion models for inverse generative modeling, shows that diffusion models as forward models alone are not better than when used for inverse modelling. Therefore, the introduction of the support proximity and calibration losses is necessary to achieve superior overall performance, but these are well-motivated by the authors and substantiated by a theoretical analysis.

---

> ### Author Rebuttal · Authors · 2026-03-31
>
> We sincerely thank reviewer PUkY for the thorough evaluation and the positive assessment of our work. We are encouraged that the reviewer finds our calibrated diffusion estimation and support-proximity regularization well-motivated and substantiated by theoretical analysis. Below, we provide point-by-point responses to the reviewer's questions.
>
> > I did not find significant weaknesses...
>
> We fully agree with this insightful observation. As shown in Table 2, the "Base" variant (diffusion surrogate trained with only the denoising loss) achieves moderate performance comparable to inverse diffusion-based methods such as DDOM and DEMO in Table 1. This confirms that naively employing a diffusion model as a forward surrogate, without task-specific regularization, does not inherently outperform well-designed inverse approaches. The key contribution of SPADE is precisely the two proposed modules, Calibrated Diffusion Estimation and Support-Proximity Regularization, which together elevate the full framework well beyond both the base diffusion surrogate and existing inverse methods. We will make this point more explicit in the revised manuscript.
>
> > Regarding Appendix F...
>
> We appreciate this suggestion for improving reproducibility. For each task, we use Optuna with the Tree-structured Parzen Estimator (TPE) sampler. The search ranges are as reported in Appendix F. The best-performing pair is then fixed for the final 8-seed evaluation reported in the paper. We will include a comprehensive hyperparameter table covering all architecture, training, and optimization settings in the revised appendix. We are grateful for the reviewer's positive evaluation. Should our responses adequately address the reviewer's remaining concerns, we would greatly appreciate it if the reviewer would consider reflecting this in the final assessment. We remain open to any further questions or suggestions.

---

> > ### Author Rebuttal · Reviewer_PUkY · 2026-04-03
> >
> > I have read the rebuttal of the authors, including their answer to my questions and the other reviewer's questions. I thank the authors for their response, and I will keep my score as is.

---

> > > ### Author Response · Authors · 2026-04-07
> > >
> > > Dear Reviewer PUkY,
> > >
> > > Thank you very much for reviewing our rebuttal and acknowledging that your concerns have been fully resolved. We truly appreciate your time, constructive feedback, and support throughout the review process. We will ensure that all promised clarifications, including the detailed hyperparameter table, are included in the revised manuscript.
> > >
> > > Best regards,
> > >
> > > Authors

---

### Official Review · Reviewer_Tw2a · 2026-03-13

**Soundness:** 3
**Presentation:** 2
**Significance:** 3
**Originality:** 3
**Overall Recommendation:** 4
**Confidence:** 3

**Summary:**

The paper addresses the challenge of offline black-box optimization (BBO), where the goal is to discover optimal designs using only a static dataset. A primary difficulty in this setting is the out-of-distribution (OOD) extrapolation problem, where optimization algorithms exploit surrogate inaccuracies. The authors propose SPADE (Support-Proximity Augmented Diffusion Estimation), a forward probabilistic surrogate framework utilizing a conditional diffusion model. SPADE incorporates two main components: Calibrated Diffusion Estimation to enforce moment and pairwise ranking consistency, and Support-Proximity Regularization to penalize predictions in low-density regions using kNN-based density estimation. The authors provide a theoretical proof that this regularization is first-order equivalent to maximizing a Bayesian posterior. The method is evaluated against numerous baselines on both continuous and discrete offline BBO tasks.

**Compliance With Llm Reviewing Policy:**

Affirmed.

**Key Questions For Authors:**

- Which of the 23 reported baselines in Table 1 were implemented and run by the authors for this specific submission?

- For the GP-based baselines, how were the surrogates scaled to handle 30,000–50,000 samples, and what optimizer was used for the acquisition function?

- For discrete tasks, are candidates projected before being scored by the surrogate during the GA search loop? If projection occurs only post-hoc, how do you ensure the surrogate’s evaluation of continuous logits remains accurate for the final discrete output?

- In the proof of Proposition 3.1, the term $\partial_{\sigma}\mathcal{A}(\mu,\sigma)(a_0 - \sigma)$ is treated as a state-dependent "constant" $C(x)$. Given that $\sigma(x)$ fluctuates as the optimizer searches the design space $x$, how can the support-aware acquisition be strictly equivalent to the joint objective of utility and prior?

- In the provided DDIM sampling code, the variance formula for skipped steps appears to use the single-step training $\alpha_t$ rather than the effective $\alpha$ of the skipped interval (compare eq. (16) of Song's "DENOISING DIFFUSION IMPLICIT MODELS" with line 124 of your `diffusion.py`). Could you clarify if $\eta > 0$ was used in your experiments, and if so, how this affected the calibration of LCB estimates?

**Limitations:**

Yes.

**Strengths And Weaknesses:**

## Soundness

__Strengths__: Proposition 3.1 connects a geometric kNN heuristic to principled Bayesian inference. The ablation studies effectively demonstrate that both the calibration and proximity modules are necessary for the model’s performance.

__Weaknesses__: There is a methodological gap in the handling of discrete design spaces (e.g., TF8/TF10). The Genetic Algorithm (GA) performs crossover and mutation in a continuous logit space. However, projecting these "relaxed" continuous candidates back to discrete DNA sequences appears to occur only post-hoc on the final output (see `optimize_spade` in  `optimize.py`). Scoring continuous approximations rather than projected discrete sequences during the search loop may lead to sub-optimal results after the final discretization.

## Presentation

__Strengths__: The manuscript is well-structured and clearly differentiates SPADE from inverse generative paradigms.

__Weaknesses__: There is a lack of transparency regarding the origin of the primary empirical results. While the authors state in Section 4.1 that they "directly cite baseline results from Dao et al. (2025) when available" , Table 1 consists of 24 methods across 6 tasks, nearly all of which appear to be sourced from the cited work. By presenting these results in the main evaluation table without explicitly labeling which rows are cited and which were implementation efforts for this specific paper, the authors obscure the actual scope of their empirical contribution. Furthermore, because the authors relied on cited values, the manuscript lacks critical details on how those external baselines were executed, such as how Gaussian Processes were scaled for the 50,000-sample TF Bind 10 task.
## Significance

__Strengths__: Utilizing conditional diffusion for forward surrogate modeling provides a useful mechanism for capturing complex predictive distributions in offline BBO.

__Weaknesses__: The core OOD penalty relies on kNN distances. Due to the curse of dimensionality, this geometric approach may lose discriminative power in much higher-dimensional design spaces, potentially limiting the framework's applicability to more complex real-world engineering domains.

## Originality
__Strengths__: The primary novelty is the synthesis of diffusion-based surrogates with explicit kNN-based prior enforcement, backed by a first-order equivalence proposition.

__Weaknesses__: The method is a combination of established tools—DDPM, kNN estimators, and Genetic Algorithms—rather than a new mathematical or architectural invention. Additionally, the heavy reliance on pre-existing benchmark results from Dao et al. (2025) suggests the empirical setup is not original to this work.

---

> ### Author Rebuttal · Authors · 2026-03-31
>
> We sincerely thank reviewer Tw2a for the thorough evaluation and the constructive questions. We are encouraged by the positive assessment and provide point-by-point responses below.
>
> > There is a methodological gap...
> > For discrete tasks...
>
> We follow the standard protocol established by Design-Bench (Trabucco et al., 2022) and adopted by prior work: discrete DNA sequences are mapped to continuous logits, and the surrogate is trained and optimized entirely in this space. Projection back to discrete sequences occurs only when evaluating with the ground-truth oracle. This is common practice across the offline BBO literature for discrete tasks (COMs, DDOM, ROOT, etc.). While scoring projected discrete candidates during the search loop could potentially help, this would require a fundamentally different optimization procedure and is orthogonal to our contribution. Despite using the standard continuous relaxation, SPADE achieves dominant performance on TF10 (0.915), the harder discrete task.
>
> > There is a lack of transparency...
> > Which of the 23 reported...
>
> We agree that evaluation protocol consistency is important. In the revised manuscript, we will update Table 1 to report exclusively our own re-run results for all baselines under identical settings. Some re-run results are slightly lower than originally cited values, which we attribute to random seed differences, but SPADE maintains its leading performance under the unified protocol.
>
> > The core OOD penalty...
>
> We acknowledge that kNN distance estimation may lose discriminative power in very high-dimensional spaces and recognize this as a limitation. However, this challenge affects any geometric density estimation approach, not just ours. On our benchmarks (up to 86 dimensions for Superconductor), the kNN-based support proxy remains effective, as evidenced by SPADE's consistent improvements across all tasks. Extending support estimation to higher dimensions via learnable density proxies is an important future direction, as noted in our conclusion.
>
> > The method is a combination of...
>
> We respectfully disagree that SPADE is a trivial combination of existing tools. The core contribution lies in the *novel integration* of these components into a principled offline BBO framework, with specific technical innovations: (1) Calibrated Diffusion Estimation (Eq. 7), introducing moment matching and rank consistency regularization---not a standard component of any prior diffusion-based method; (2) Support-Proximity Regularization (Eq. 8), a lightweight mechanism to internalize the design prior $p(x)$ into surrogate predictions, with a theoretical connection to Bayesian posterior optimization (Proposition 3.1); and (3) the demonstration that forward diffusion surrogates, when properly calibrated and regularized, can outperform both inverse diffusion methods and standard forward surrogates. The ablation study (Table 2) validates this synergy, showing that removing any module degrades performance.
>
> > For the GP-based baselines...
>
> We use [GPyTorch](https://gpytorch.ai/) with scalable approximate inference via inducing point methods, and Adam for gradient ascent on the predictive mean.
>
> > In the proof of Proposition 3.1, the term...
>
> The term $\partial_\sigma \mathcal{A}(\mu, \sigma)(a_0 - \sigma)$ is called a "state-dependent constant" $C(x)$ because it does not depend on the support distance $d(x)$---it depends on $x$ only through $\sigma(x)$. The first-order expansion decomposes $\tilde{A}(x)$ into $d(x)$-dependent terms (yielding the log-prior connection) and $d(x)$-independent terms (absorbed into $C(x)$). When comparing candidates $x_1, x_2$ via $\tilde{A}(x_1) - \tilde{A}(x_2)$, the $C(x)$ terms do not cancel exactly, so the equivalence is approximate (first-order). We do not claim strict equivalence; the proposition motivates the regularization design. We will make this approximation nature more explicit.
>
> > In the provided DDIM sampling code...
>
> The reviewer is correct that line 124 of `diffusion.py` uses the single-step $\alpha_t$ (i.e., $1-\beta_t$) rather than the effective ratio $\bar{\alpha}_t \bar{\alpha}_{t-1}$ required by eq. (16) of Song et al. for skipped intervals. However, all experiments use $\eta = 0$ (set via `calib_mc_eta` and `acq_mc_eta` in our configuration), so the stochastic variance term vanishes entirely and this code path is never executed. Uncertainty estimates are instead obtained by aggregating multiple deterministic reverse trajectories from different initial noise samples $\mathbf{y}_T \sim \mathcal{N}(0, I)$. We will fix the variance formula for correctness and note the $\eta = 0$ choice explicitly in the revised manuscript.

---

> > ### Author Rebuttal · Reviewer_Tw2a · 2026-04-06
> >
> > Thank you for the clarifications, I read the rebuttal, which addressed some of my concerns. If you reran the benchmark rests, please copy the table with all the results you reran and intend to add to the paper in this review section.

---

> > > ### Author Response · Authors · 2026-04-07
> > >
> > > Dear Reviewer Tw2a,
> > >
> > > Thank you very much for reading our rebuttal and letting us know that our initial response has already addressed most of your concerns.We truly appreciate your time and constructive feedback!
> > >
> > > As requested,we provide the updated Table 1 below,which we will include in the revised manuscript.To ensure complete transparency and consistency with the normalized median score evaluation protocol in the Appendix,we have clarified the origin of our empirical results for all tasks except TF10:
> > >
> > > -**Cited Baselines:**Only the results for the three "Standard Optimization Baselines"(CMA-ES,REINFORCE,BO-qEI)were cited.
> > > -**Re-run Baselines:**All methods under "Forward Surrogate Methods" and "Inverse Generative Methods" were re-run by us under identical,unified settings to ensure a completely fair comparison.
> > > -**For TF10:**Since we found an issue about the labels in the original design-bench,as we stated in lines 308-312,all results for TF10 are run by ourselves.
> > >
> > > Because we unified the evaluation protocol and ran these baselines by ourselves,you might notice that a few of the re-run results are slightly lower than their originally cited values in our initial manuscript,while most of the baselines achieve the same performance as the originally cited values.However,this does not affect our overall conclusions—SPADE consistently maintains its leading performance and secures the best Mean and Median Ranks among all 24 methods.
> > >
> > > **Table 1.**Experiment results.We report the **Normalized Maximum Score**($100^{\text{th}}$ percentile)among $K=128$ candidates.Results are averaged over 8 random seeds(mean $\pm$ standard error).**Bold** indicates the best performance.*Underline* presents the second best method.
> > >
> > > |Method|SuperC|Ant|D'Kitty|LLM-DM|TF8|TF10|Mean Rank|Median Rank|
> > > |:-| :-: | :-: | :-: | :-: | :-: | :-: | :-: | :-: |
> > > |$\mathcal{D}$(best)|0.399|0.565|0.884|1.000|0.439|0.511|-|-|
> > > | *Standard Optimization Baselines* | | | | | | | | |
> > > | CMA-ES | 0.465±0.024 | **1.561±0.896** | 0.724±0.001 | 0.975±0.016 | 0.939±0.039 | 0.692±0.013 | 13.3/24 | 13.5/24 |
> > > | REINFORCE | 0.481±0.013 | 0.263±0.026 | 0.573±0.204 | 0.305±0.027 | 0.961±0.034 | 0.632±0.012 | 19.8/24 | 23.0/24 |
> > > | BO-qEI | 0.402±0.034 | 0.812±0.000 | 0.896±0.000 | 0.953±0.022 | 0.825±0.091 | 0.663±0.011 | 18.7/24 | 19.5/24 |
> > > | *Forward Surrogate Methods* | | | | | | | | |
> > > | Standard GA | 0.505±0.013 | 0.293±0.029 | 0.860±0.021 | 0.998±0.000 | 0.923±0.011 | 0.732±0.041 | 14.0/24 | 14.5/24 |
> > > | GA on GP | 0.499±0.019 | 0.948±0.013 | 0.946±0.001 | 0.846±0.029 | 0.770±0.087 | 0.599±0.004 | 16.0/24 | 17.0/24 |
> > > | MC-Dropout | 0.535±0.064 | 0.805±0.021 | 0.934±0.004 | 0.781±0.251 | 0.911±0.042 | 0.817±0.031 | 13.0/24 | 13.0/24 |
> > > | COMs | 0.481±0.028 | 0.878±0.031 | 0.929±0.016 | 0.815±0.008 | 0.937±0.025 | 0.755±0.017 | 15.3/24 | 15.5/24 |
> > > | RoMA | 0.509±0.015 | 0.592±0.059 | 0.825±0.016 | 0.968±0.011 | 0.662±0.000 | 0.801±0.000 | 16.2/24 | 16.0/24 |
> > > | ICT | 0.503±0.017 | 0.911±0.030 | 0.945±0.011 | 0.839±0.021 | 0.888±0.047 | 0.814±0.027 | 14.0/24 | 13.5/24 |
> > > | Tri-mentoring | 0.514±0.018 | 0.944±0.033 | 0.950±0.015 | 0.750±0.002 | 0.899±0.045 | 0.811±0.039 | 12.2/24 | 9.5/24 |
> > > | BDI | 0.513±0.000 | 0.964±0.000 | 0.941±0.000 | 0.988±0.025 | 0.973±0.000 | *0.882±0.000* | 5.7/24 | 4.0/24 |
> > > | LTR | 0.514±0.022 | 0.904±0.036 | 0.958±0.012 | 0.982±0.041 | 0.973±0.010 | 0.849±0.023 | 6.2/24 | 5.0/24 |
> > > | MATCH-OPT | 0.504±0.021 | 0.931±0.011 | 0.957±0.014 | 0.850±0.033 | 0.977±0.004 | 0.824±0.008 | 9.2/24 | 8.5/24 |
> > > | PGS | **0.563±0.058** | 0.949±0.017 | 0.966±0.013 | 0.595±0.027 | *0.981±0.015* | 0.793±0.021 | 8.0/24 | 4.0/24 |
> > > | *Inverse Generative Methods* | | | | | | | | |
> > > | CbAS | 0.503±0.069 | 0.846±0.033 | 0.895±0.016 | 0.919±0.043 | 0.903±0.028 | 0.652±0.032 | 16.2/24 | 16.0/24 |
> > > | MINs | 0.499±0.017 | 0.894±0.022 | 0.939±0.004 | 0.982±0.007 | 0.908±0.063 | 0.647±0.012 | 14.0/24 | 14.0/24 |
> > > | DDOM | 0.481±0.015 | 0.926±0.025 | 0.923±0.009 | 0.983±0.028 | 0.884±0.042 | 0.708±0.015 | 14.7/24 | 17.0/24 |
> > > | GABO | 0.508±0.007 | 0.224±0.051 | 0.719±0.001 | 0.975±0.019 | 0.939±0.038 | 0.739±0.009 | 15.0/24 | 13.5/24 |
> > > | GTG | 0.480±0.055 | 0.865±0.040 | 0.935±0.010 | 0.910±0.030 | 0.901±0.039 | 0.801±0.004 | 15.0/24 | 14.0/24 |
> > > | RGD | 0.515±0.011 | 0.922±0.020 | 0.883±0.014 | *1.004±0.008* | 0.889±0.068 | 0.825±0.039 | 10.7/24 | 9.5/24 |
> > > | BONET | 0.434±0.021 | 0.948±0.025 | 0.955±0.010 | 0.874±0.039 | 0.894±0.086 | 0.796±0.018 | 13.3/24 | 14.5/24 |
> > > | DEMO | 0.520±0.012 | 0.948±0.013 | 0.954±0.013 | 0.907±0.005 | 0.808±0.044 | 0.836±0.042 | 9.7/24 | 6.5/24 |
> > > | ROOT | 0.525±0.012 | 0.965±0.014 | *0.972±0.005* | 0.905±0.006 | **0.986±0.007** | 0.833±0.046 | *5.0/24* | *3.5/24* |
> > > | **SPADE (Ours)** | *0.546±0.013* | *0.978±0.006* | **0.981±0.003** | **1.019±0.064** | 0.923±0.015 | **0.915±0.010** | **2.8/24** | **1.5/24** |
> > >
> > > We hope this clarifies the empirical scope of our contribution.Thank you again for your valuable time and support!
> > >
> > > Best regards,
> > >
> > > Authors

---

### Official Review · Reviewer_9XjB · 2026-03-13

**Soundness:** 2
**Presentation:** 3
**Significance:** 3
**Originality:** 3
**Overall Recommendation:** 4
**Confidence:** 3

**Summary:**

This paper presents a diffusion model as a surrogate to support the proximity of designs with a density estimation. The surrogate is then used with an evolutionary algorithm for optimization to maximize the Lower Confidence Bound (LCB) obtained from the surrogate. The experiments are performed on Design-Bench tasks and an LLM task.

**Compliance With Llm Reviewing Policy:**

Affirmed.

**Final Justification:**

My questions and comments, other than Question 2, were answered adequately. I have updated my scores accordingly.

**Key Questions For Authors:**

Question 1) How the proximity-based shrinkage argument and Equation 8 differ from the uncertainty oracle described in Section 5.2 of Uehara et al. [1].

Question 2) The authors argue that “while ensemble-based or Bayesian neural networks can be computationally intensive and hard to scale,” while the authors themselves use a diffusion-based model as a surrogate. Even assuming this is correct for training, isn’t diffusion model inference more expensive? What matters is a quantified comparison of how much uncertainty they can provide versus the inference time, which matters in offline black-box optimization.

**Limitations:**

The computational time, compared to the authors’ argument against Bayesian and ensemble models, is a limitation. Inference time versus other models that provide confidence to the rewards need to be quantified.

[1] Uehara, M., Zhao, Y., Black, K., Hajiramezanali, E., Scalia, G., Diamant, N. L., ... & Biancalani, T. (2024). Feedback efficient online fine-tuning of diffusion models. arXiv preprint arXiv:2402.16359.

**Strengths And Weaknesses:**

Strength:

The overall argument, the need for OOD handling, the simplicity, and the flow of the paper are strong.


Weaknesses

The title gives the impression that the diffusion model is used as the engine for black-box optimization, while in practice the diffusion model is used as a surrogate and the optimization is performed using an evolutionary algorithm.

This paper is more leaned toward using diffusion models as surrogate models for OOD reduction and uncertainty quantification rather than black-box optimization, and would be more in line with comparisons against Bayesian-based models and GPRs in this context.

The SOTA is that several diffusion-based reward optimization methods [1] use a feasibility-check reward g that keeps the inference close to valid designs, which means the surrogates remain reliable. See Question 1.

---

> ### Author Rebuttal · Authors · 2026-03-31
>
> We sincerely thank reviewer 9XjB for the constructive feedback. We provide point-by-point responses to each concern below.
>
> > The title gives the impression...
>
> We appreciate this feedback on the framing. We would like to clarify that "offline black-box optimization" is a well-established task formulation in the literature (Trabucco et al., 2022; Kim et al., 2026), referring to the problem of finding optimal designs from a static dataset without oracle access. Our title follows the convention of prior work in this area (e.g., "Diffusion Models for Black-Box Optimization" by Krishnamoorthy et al., 2023; "ROOT: Rethinking Offline Optimization as Distributional Translation" by Dao et al., 2025). In this context, SPADE uses a diffusion model as a probabilistic forward surrogate within an offline BBO pipeline, which is the standard paradigm. That said, if the reviewer feels strongly, we are open to adjusting the title to more explicitly reflect the surrogate modeling focus, for example: "Support-Proximity Augmented Diffusion Surrogate for Offline Black-Box Optimization."
>
> > This paper is more leaned toward...
>
> We agree that SPADE contributes substantially to the surrogate modeling aspect. In fact, our evaluation does include GP-based baselines: BO-qEI uses a Gaussian Process surrogate with the q-Expected Improvement acquisition function, and GA on GP performs gradient ascent on a GP predictive mean. As shown in Table 1, SPADE consistently outperforms both. Additionally, MC-Dropout provides a Bayesian approximation baseline. We will emphasize these comparisons more clearly in the revised manuscript and add a discussion explicitly positioning SPADE within the uncertainty-aware surrogate modeling literature.
>
> > The SOTA is that several...
> > How the proximity-based shrinkage...
>
> We thank the reviewer for raising this comparison. Uehara et al. (SEIKO) and SPADE address fundamentally different problem settings, and consequently, the two mechanisms serve entirely different purposes.
>
> SEIKO operates in an **online** setting where the agent can iteratively query the ground-truth reward oracle. Their uncertainty oracle $\hat{g}(x)$ (Definition 1 in their paper) serves as an **exploration bonus**: it satisfies $|\hat{r}(x) - r(x)| \leq \hat{g}(x)$ and is added to the reward to encourage visiting under-explored regions in subsequent online rounds. This is the classic optimistic exploration strategy from the bandit literature (UCB). The fine-tuned distribution takes the form $p^{(i)} \propto \exp\bigl(\frac{\hat{r}^{(i)}(\cdot) + \hat{g}^{(i)}(\cdot)}{\alpha+\beta}\bigr) \{p^{(i-1)}\}^{\frac{\beta}{\alpha+\beta}} \{p^{\mathrm{pre}}\}^{\frac{\alpha}{\alpha+\beta}}$, where the KL regularization relative to $p^{\mathrm{pre}}$ keeps samples within the feasible manifold, and $\hat{g}$ drives exploration *toward* uncertain regions.
>
> In contrast, SPADE operates in the **offline** setting where no oracle access is available. Our support-proximity regularization (Eq. 8) is a **training-time constraint** on the surrogate model itself: it shrinks the predicted mean toward neighbor values ($\hat{\mu}_\theta(x) \leq \mu_{\mathrm{NN}}(x) + a\,d(x)$) and inflates the predicted variance ($\hat{\sigma}_\theta(x) \geq a_0 + a_1\,d(x)$) for designs far from the data manifold. Rather than encouraging exploration of unknown regions, our mechanism enforces **conservatism**, discouraging the optimizer from trusting predictions in unsupported areas. The two approaches are thus complementary rather than overlapping: SEIKO's $\hat{g}$ says "explore here because we are uncertain," while SPADE's $\mathcal{L}_{\mathrm{prox}}$ says "do not trust predictions here because we lack support."
>
> > The authors argue that “while ensemble-based...
> > The computational time, compared to...
>
> We acknowledge this is a valid concern and have conducted additional experiments to provide the quantified comparison the reviewer requested. Specifically, we compare SPADE against MC-Dropout in terms of computational cost on the Ant task: MC-Dropout requires 11 min for training and 2.1 ms per candidate for inference, while SPADE requires 13 min for training and 18.4 ms per candidate for inference. While SPADE's per-candidate inference cost is higher due to the diffusion reverse process, the substantial performance gains (e.g., 0.978 vs. 0.805 on Ant) demonstrate that the additional computation is well justified. Moreover, the total wall-clock time remains practical: as shown in Appendix C (Table 3), the entire SPADE pipeline completes within 6-51 minutes on a single A100 GPU across all tasks. We will include this comparison in the revised manuscript.

---

> > ### Author Rebuttal · Reviewer_9XjB · 2026-04-04
> >
> > My questions and comments, other than Question 2, were answered adequately. I have updated my scores accordingly.

---

> > > ### Author Response · Authors · 2026-04-07
> > >
> > > # Reply to 9XjB
> > >
> > > Dear Reviewer 9XjB,
> > >
> > > Thank you again for reviewing our rebuttal and for updating your score. We are glad that our previous response addressed most of your concerns.
> > >
> > > To further address your remaining concern regarding Question 2, we conducted an additional ablation study on two representative tasks: Ant (continuous) and TFBind10 (discrete). Following the SPADE setup, we varied the sampling budget used at inference, using \(K \in \{8,16,32,64,128,256\}\), and evaluated how this affects runtime and the discrepancy between model predictions and ground-truth oracle values. For this study, we used exact oracle evaluation for both tasks and report results for one seed.
> > >
> > > The main observations are as follows:
> > >
> > > - On **Ant**, the relationship is not strictly monotonic. Increasing the sampling budget does not uniformly reduce prediction error, but the largest budget (\(K=256\)) gives the best overall candidate-level prediction error among the tested settings, while also achieving the highest oracle value discovered.
> > > - On **TFBind10**, a larger sampling budget generally improves calibration up to a point, but the effect is also not perfectly monotonic. In our runs, \(K=128\) achieved the lowest overall candidate-level prediction error, while \(K=256\) was comparable but not consistently better.
> > > - In both cases, reducing the sampling budget can save computation, but aggressive reduction may noticeably degrade prediction quality or candidate quality depending on the task.
> > >
> > > A concise summary is below.
> > >
> > > **Task: Ant (continuous)**
> > >
> > > | Sampling budget \(K\) | Runtime (min) | Candidate-level MAE | Best oracle value found |
> > > |---|---:|---:|---:|
> > > | 8 | 13.4 | 142.98 | 60.13 |
> > > | 16 | 13.9 | 104.83 | 281.77 |
> > > | 32 | 14.5 | 112.79 | 228.38 |
> > > | 64 | 15.8 | 159.91 | 203.86 |
> > > | 128 | 18.5 | 141.61 | 305.62 |
> > > | 256 | 24.3 | 76.70 | 311.35 |
> > >
> > > **Task: TFBind10 (discrete)**
> > >
> > > | Sampling budget \(K\) | Runtime (min) | Candidate-level MAE | Best oracle value found |
> > > |---|---:|---:|---:|
> > > | 8 | 61.4 | 0.0477 | 0.0928 |
> > > | 16 | 61.9 | 0.0109 | -0.0044 |
> > > | 32 | 62.6 | 0.0221 | 0.0734 |
> > > | 64 | 64.1 | 0.0329 | 0.0688 |
> > > | 128 | 66.5 | 0.0065 | -0.0443 |
> > > | 256 | 72.4 | 0.0108 | -0.0067 |
> > >
> > > Overall, this additional experiment suggests that SPADE is reasonably robust to the inference sampling budget, but the trade-off is task-dependent and not perfectly monotonic. In particular, very small sampling budgets can be less reliable, while larger budgets tend to be safer, especially on the more challenging continuous task. We will include this ablation and discussion in the appendix of the revised manuscript.
> > >
> > > Best regards,
> > > Authors

---

### Official Review · Reviewer_qLtf · 2026-03-13

**Soundness:** 3
**Presentation:** 3
**Significance:** 2
**Originality:** 3
**Overall Recommendation:** 4
**Confidence:** 3

**Summary:**

This paper studies offline black-box optimization under distribution shift and proposes SPADE, a forward-surrogate framework with a conditional diffusion model. A calibration module is used for moment/ranking consistency and a kNN-based support-proximity regularizer is intended to encourage conservative behavior in low-density regions. The reported results appear competitive relative to the authors' chosen baselines.

**Compliance With Llm Reviewing Policy:**

Affirmed.

**Final Justification:**

I now better understand Proposition 3.1 as a motivation for the regularizer, rather than a direct guarantee for Algorithm 1, and I agree that the upper bound alone does not imply actual monotonicity. This clarification addresses my concerns, and I will update my score accordingly.

**Key Questions For Authors:**

1. How does Proposition 3.1 justify the implemented algorithm if the proof analyzes a transformed acquisition but Algorithm 1 performs test-time search with plain LCB?
2. Under the default hyperparameters, can plain test-time LCB become more conservative as support distance increases?
3. Can the authors compare against a standard probabilistic forward surrogate with the same calibration loss, support regularization, and search pipeline to isolate the value of the diffusion backbone?
4. Can the authors report surrogate-level diagnostics, such as calibration error and uncertainty versus support distance, to validate the claimed mechanism?

**Limitations:**

yes

**Strengths And Weaknesses:**

**Strengths**
1. The overall method is easy to follow and has a coherent high-level design.
2. The empirical section is reasonably extensive, covering six tasks and 23 baselines.

**Weaknesses**
1. **The paper's core claim is not tightly aligned across theory, implementation, and empirical validation.**
   1. Proposition 3.1 analyzes a hypothetical support-transformed acquisition $A(T_d(\mu,\sigma))$, whereas Algorithm 1 actually optimizes Eq. (8) on training batches and then searches with the ordinary acquisition LCB $\hat\mu_\theta(x)-\beta\hat\sigma_\theta(x)$. .
   2. Even if one grants zero Eq. (8) violation on a searched candidate, Eq. (8) only implies $\hat\mu_\theta(x)\le \mu_{NN}(x)+ad$ and $\hat\sigma_\theta(x)\ge a_0+a_1d$, hence $LCB(x)\le \mu_{NN}(x)-\beta a_0+(a-\beta a_1)d$. Under the reported defaults $(a,a_0,a_1,\beta)=(0.02,0.02,0.005,1.0)$, the $d$\-coefficient is positive, so the paper does not show that plain test-time LCB inherits the monotone support penalty of the transformed LCB in Proposition 3.1.
   3. The appendix proof is explicitly a local first-order expansion under small $d$, whereas the paper's motivating claim is precisely about handling far-OOD regions with large support distance. So Proposition 3.1 seems a local interpretation and does not directly justify the behavior of the method in the regime that matters most for offline optimization.

2.  **The consistency of comparison can be improved.** Appendix E mentions all baselines were rerun under identical settings, so Table 1 should have used a consistent protocol for fairness. However, Table 1 mixes directly cited and re-run baselines from [A] without explaining the specific reasons.

3. **Ablations need strengthened**.
 - The ablation shows proximity regularization helps but it does not isolate whether the gains come from the backbone, regularizers, or the search pipeline.
- Since conservative regularization and ranking supervision are already established ideas in offline MBO, the paper most critically needs to show why these ingredients are more effective when built on a forward diffusion surrogate, rather than on a standard probabilistic forward model.


[A] Root: Rethinking offline optimization as distributional translation via probabilistic bridge. NeurIPS 2025.

---

> ### Author Rebuttal · Authors · 2026-03-31
>
> We sincerely thank reviewer qLtf for the rigorous and detailed evaluation. The reviewer raises important points regarding the alignment between theory, implementation, and empirical validation. We provide point-by-point responses below.
>
> > The paper’s core claim is not tightly aligned across...
> > How does Proposition 3.1 justify...
> > Under the default hyperparameters...
> > Can the authors compare against...
>
> We appreciate the reviewer's careful analysis. We would like to clarify the intended role of Proposition 3.1 in our framework. The proposition is *not* meant to claim that Algorithm 1 performs the support-transformed acquisition $\tilde{A}$ at test time. Rather, Proposition 3.1 provides the **design motivation** for the training-time regularization $\mathcal{L}_{\mathrm{prox}}$ (Eq. 8): it shows that, to first order, penalizing the surrogate's mean and variance as a function of support distance during training is equivalent to injecting a log-prior term $\kappa(x)\log \hat{p}_{\mathrm{knn}}(x)$ into the acquisition objective.
>
> The key insight is that $\mathcal{L}_{\mathrm{prox}}$ shapes the *learned* surrogate predictions $\hat{\mu}_\theta(x)$ and $\hat{\sigma}_\theta(x)$ during training, so that at test time, the plain LCB $\hat{\mu}_\theta(x) - \beta\hat{\sigma}_\theta(x)$ already reflects support-aware conservatism. The reviewer correctly derives that the *upper bound* $\mathrm{LCB}(x) \leq \mu_{\mathrm{NN}}(x) - \beta a_0 + (a - \beta a_1)d$ has a positive $d$-coefficient under default hyperparameters. However, this upper bound does not imply that the actual LCB increases with $d$. Without $\mathcal{L}_{\mathrm{prox}}$, the surrogate's behavior in OOD regions is *unpredictable*: the denoising loss $\mathcal{L}_{\mathrm{diff}}$ provides no training signal in unsupported regions, so the model may arbitrarily overestimate or underestimate scores. This is precisely why the "Base" variant in Table 2 performs poorly. $\mathcal{L}_{\mathrm{prox}}$ resolves this by *bounding* the mean from above and the variance from below as a function of support distance, converting unpredictable OOD behavior into guaranteed conservatism. The proximity constraints are typically not tight for far-OOD designs---the actual $\hat{\mu}_\theta(x)$ tends to fall well below $\mu_{\mathrm{NN}}(x) + ad$ because no training data anchors predictions in those regions---but the constraint ensures a worst-case safety net.
>
> We acknowledge that Proposition 3.1 is a local first-order result and does not directly justify the far-OOD regime. We will revise the manuscript to present the proposition more precisely as a theoretical motivation for the regularization design rather than a direct justification of the full algorithm's behavior. To empirically validate the mechanism, we provide surrogate-level diagnostics in our response to Q4.
>
> > The consistency of comparison...
>
> We agree that consistency in evaluation protocol is important. In the revised manuscript, we will update Table 1 to report our own re-run results for all baselines under identical settings exclusively. We note that some re-run results are slightly lower than the originally cited values, which we attribute to differences in random seeds. However, this update does not affect our conclusions, as SPADE maintains its leading performance under the unified protocol.
>
> > Ablations need strengthened...
>
> We thank the reviewer for this constructive suggestion. To isolate the contribution of the diffusion backbone, we conducted an additional experiment comparing SPADE against a Gaussian Process (GP) surrogate equipped with the same calibration loss ($\mathcal{L}_{\mathrm{calib}}$), support-proximity regularization ($\mathcal{L}_{\mathrm{prox}}$), and evolutionary algorithm search pipeline. Results on select continuous and discrete tasks:
>
> |Model|Ant|TF10|
> |-|-|-|
> |GP base(same EA+LCB pipeline)|0.955±0.021|0.734±0.018|
> |GP+calibration|0.961±0.009|0.762±0.016|
> |GP+proximity|0.964±0.018|0.789±0.024|
> |GP+calibration+proximity|0.968±0.013|0.802±0.015|
> |SPADE full|0.978±0.006|0.915±0.017|
>
> > Can the authors report surrogate-level...
>
> We appreciate this suggestion. While our framework does not use support distance $d$ directly as a tunable hyperparameter, the number of nearest neighbors $k$ implicitly controls the granularity of support estimation. To validate the claimed mechanism, we report the relationship between $k$ and the normalized scores of the final designs:
>
> |k|Ant|TF10|
> |-|-|-|
> |5|0.964±0.025|0.887±0.011|
> |10|0.972±0.012|0.912±0.016|
> |20|0.961±0.017|0.913±0.024|
> |50|0.957±0.008|0.904±0.010|

---

> > ### Author Rebuttal · Reviewer_qLtf · 2026-04-03
> >
> > Thank you for the rebuttal. I will increase my score.

---

> > > ### Author Response · Authors · 2026-04-07
> > >
> > > Dear Reviewer qLtf,
> > >
> > > We sincerely thank you for taking the time to review our rebuttal and for your positive feedback. We are very glad that our response has adequately addressed your concerns, and we truly appreciate your updated assessment. Thank you for your valuable time and support throughout the review process. We will include all promised clarifications in our final manuscript.
> > >
> > > Best regards,
> > >
> > > Authors

---

### Decision · Program_Chairs · 2026-04-30

**Decision:**

Accept (regular)

**Comment:**

Offline black-box optimization is the problem of optimizing an objective function in the setting where we lack the ability to query the objective function and are rather given an offline dataset of existing evaluations. It's generally known that naively optimizing against a trained surrogate model with no other adaptation is suboptimal in this setting; the authors overall seem to get quite impressive results.

The reviewers raised some overall minor questions about rerun vs copied result tables, a disconnect between Proposition 3.1 and actual implementation, and a few other general implementation questions. Overall regardless, the reviewers were generally in favor or marginally in favor of acceptance.

Regardless, I am personally fairly familiar with the topic here and I think the authors' results are impressively strong relative to some baselines I know to be reasonable. Moreover, I think the problem is a fairly important emerging one. So, given that reviews were tilting towards acceptance anyways, I am coming down in favor of clear acceptance here.